# Flat bands, non-trivial band topology and rotation symmetry breaking in layered kagome-lattice RbTi$_3$Bi$_5$

Zhicheng Jiang[1,2,7], Zhengtai Liu [1,3,7] ✉, Haiyang Ma[4,5,7], Wei Xia[4,5,7], Zhonghao Liu [1], Jishan Liu[1,3], Soohyun Cho [1], Yichen Yang[1], Jianyang Ding[1], Jiayu Liu[1], Zhe Huang[4], Yuxi Qiao[1], Jiajia Shen[1], Wenchuan Jing[1], Xiangqi Liu[5,6], Jianpeng Liu [4,5] ✉, Yanfeng Guo [4,5] ✉ & Dawei Shen [1,6] ✉

A representative class of kagome materials, AV$_3$Sb$_5$ (A = K, Rb, Cs), hosts several unconventional phases such as superconductivity, $\mathbb{Z}_2$ non-trivial topological states, and electronic nematic states. These can often coexist with intertwined charge-density wave states. Recently, the discovery of the iso-structural titanium-based single-crystals, ATi$_3$Bi$_5$ (A = K, Rb, Cs), which exhibit similar multiple exotic states but without the concomitant charge-density wave, has opened an opportunity to disentangle these complex states in kagome lattices. Here, we combine high-resolution angle-resolved photo-emission spectroscopy and first-principles calculations to investigate the low-lying electronic structure of RbTi$_3$Bi$_5$. We demonstrate the coexistence of flat bands and several non-trivial states, including type-II Dirac nodal lines and $\mathbb{Z}_2$ non-trivial topological surface states. Our findings also provide evidence for rotational symmetry breaking in RbTi$_3$Bi$_5$, suggesting a directionality to the electronic structure and the possible emergence of pure electronic nematicity in this family of kagome compounds.

In recent years, kagome-lattice materials containing frustrated corner-sharing triangles provide an exciting platform to investigate the interplay between electron correlations, lattice geometry and band topology[1–9]. Since electrons are localized in the unique frustrated honeycomb hexagonal structure, kagome-lattice materials simultaneously host flat bands, van Hove singularities (VHSs) and Dirac band crossings, which subsequently give rise to a series of exotic quantum states depending on the band filling, such as quantum spin liquid, fractional quantum Hall states and density waves[10–15]. In particular, kagome-lattice metals have long been predicted to host

unconventional chiral and spin-triplet superconductivity (SC) when the chemical potential is tuned to be close to VHSs[16–20]. Consequently, the layered kagome-lattice AV$_3$Sb$_5$ (A = K, Rb, and Cs), in which both the $\mathbb{Z}_2$ non-trivial topological band[21] and superconductivity[22,22–26] were discovered, rapidly sparked enormous research interests[19,26–36]. The suggested strong-coupling super-conductivity with possible triplet pairing and non-trivial band topology therein might pave the way for realizing the long-sought-after Majorana zero modes. Furthermore, more unexpected exotic ordered states discovered in AV$_3$Sb$_5$, including unidirectional charge

[1]State Key Laboratory of Functional Materials for Informatics, Shanghai Institute of Microsystem and Information Technology, Chinese Academy of Sciences, 200050 Shanghai, China. [2]Center of Materials Science and Optoelectronics Engineering, University of Chinese Academy of Sciences, 100049 Beijing, China. [3]Shanghai Synchrotron Radiation Facility, Shanghai Advanced Research Institute, Chinese Academy of Sciences, 201210 Shanghai, China. [4]School of Physical Science and Technology, ShanghaiTech University, 201210 Shanghai, China. [5]ShanghaiTech Laboratory for Topological Physics, ShanghaiTech University, 201210 Shanghai, China. [6]National Synchrotron Radiation Laboratory, University of Science and Technology of China, 230029 Hefei, China. [7]These authors contributed equally: Zhicheng Jiang, Zhengtai Liu, Haiyang Ma and Wei Xi. ✉e-mail: ztliu@mail.sim.ac.cn; liujp@shanghaitech.edu.cn; guoyf@shanghaitech.edu.cn; dwshen@ustc.edu.cn

order, rotation symmetry breaking and roton pair density waves[26,28,34], further stimulated the research interest in this family of materials because they are akin to those correlated physics in high-temperature superconductors. To date, it has been becoming one of the research frontiers, which aims to decode these ordered states and then further disentangle their interrelation.

Moreover, the canonical electronic structure with symmetry-protected flat and topological bands as well renders kagome-lattice materials a promising sandbox to explore and try out novel quantum states[37,38]. Unfortunately, previously reported band structures have demonstrated that both flat bands and $\mathbb{Z}_2$ topological surface states in $AV_3Sb_5$ are far away from or well above the Fermi level ($E_F$), which consequently triggers a repulsive barrier to the interaction of flat or non-trivial topological bands with exotic ordered states in the vicinity of $E_F$. Recently, the discovery of new $AB_3C_5$ members $ATi_3Bi_5$ (A = Rb, Cs), in which V ($3d^3$) and Sb atoms are completely substituted by Ti ($3d^2$) and heavier Bi atoms, respectively, compared to $AV_3Sb_5$, paves a promising way to address this issue[39,40]. On one hand, the substantial equivalent hole doping due to the reduction of band filling effectively brings the flat bands close to $E_F$[41–43]; on the other hand, the stronger spin-orbital coupling (SOC) can be more likely to cause the band inversion, leading to richer topological band characteristics. Although the superconductivity therein are still in debate, there is no doubt that $ATi_3Bi_5$ could provide a promising playground for further exploring the interplay between topology, superconductivity and geometric frustration in kagome-lattice. Furthermore, since no evidence on the existence of CDW phase occurring in $ATi_3Bi_5$ have been reported, they constitute ideal model systems for comparison study of the electronic nematicity in kagome-lattice without the interference of translation symmetry breaking CDW as in $AV_3Sb_5$.

In this work, we present a report on the low-lying electronic structure of $RbTi_3Bi_5$, one typical $ATi_3Bi_5$ compound, using high-resolution angle-resolved photoemission spectroscopy (ARPES) together with first-principles calculations. Spectroscopic signatures of multiple coexisting topological states in this sibling material of $AV_3Sb_5$ are identified in ARPES datasets by comparison with DFT predictions, including flat bands, type-II Dirac nodal lines and topological surface states in the vicinity of $E_F$. Intriguingly, we discover that the Dirac cones and flat bands are located at the same binding energy and intertwined with each other. Meanwhile, our results provide evidence for an important role of VHS around the Fermi level in the formation of CDW

in kagome metals. Furthermore, autocorrelation intensity plots from ARPES data suggest the anisotropic scattering along different wave-vector directions, indicating the rotation symmetry breaking in $RbTi_3Bi_5$ at low temperatures.

## Results

### The crystal and band structure of $RbTi_3Bi_5$

$RbTi_3Bi_5$ shares the same crystal structure as $AV_3Sb_5$, and it crystallizes in the space group P6/mmm with lattice constants $a$ = 5.77 Å and $c$ = 9.065 Å[39,40,44,45]. As shown in Fig. 1a, it consists of Ti-Bi slabs, Bi honeycomb layers and alkali Rb triangle networks stacking alternatively along the $c$ axis. Here, those core two-dimensional (2D) kagome layers are composed of Ti atoms in Ti-Bi slabs [Fig. 1b]. The three-dimensional (3D) and projected (001) surface Brillouin zones (BZs) of $RbTi_3Bi_5$ are illustrated in Fig. 1c, in which plotted black dots indicate high-symmetry momenta. More details on the sample preparation, characterization and calculations can be found in Supplementary Information (Supplementary Fig. 1).

Figure 1 d and e show calculated band structures of $RbTi_3Bi_5$ wo/w the SOC, respectively. Although the substantial SOC from Bi atoms introduces significant changes to some of bands, we can still identify typical band features of kagome-lattice, e.g., VHSs located at ~$E_F$+0.2 eV at $M/L$ and Dirac points (DPs) located at ~$E_F$+0.5 eV at $K/H$, which are highlighted in the zoomed-in circles on the right. However, distinct from $AV_3Sb_5$, both features are far away from $E_F$. In contrast, we have found pronounced flat bands, which are located just slightly below $E_F$ (~ −0.25 eV). Furthermore, along the high-symmetry direction $\Gamma/A$-$M/L$ and at nearly the same binding energy, we have discovered a type-II DP as well, as shown in the right enlarged blue circle.

### The flat bands in $RbTi_3Bi_5$

We then experimentally searched for the predicted flat bands. Firstly, we demonstrate ARPES intensity mapping of Fermi surface taken with 66 eV photons for $RbTi_3Bi_5$, as illustrated in Fig. 2a. There exist four dispersive bands crossing $E_F$ labeled by $\alpha$ to $\delta$, forming one circle-like ($\alpha$ band) and one hexagonal-like ($\beta$ band) electron pocket. The $\gamma$ band crosses through $E_F$ two times, which forms a hexagonal-like electron pocket around zone centre and a triangle-like hole pocket around the corner of BZ. Additionally, the $\delta$ band constitutes a rhombic-like hole pocket around the boundary of BZ. As the increase of binding energy,

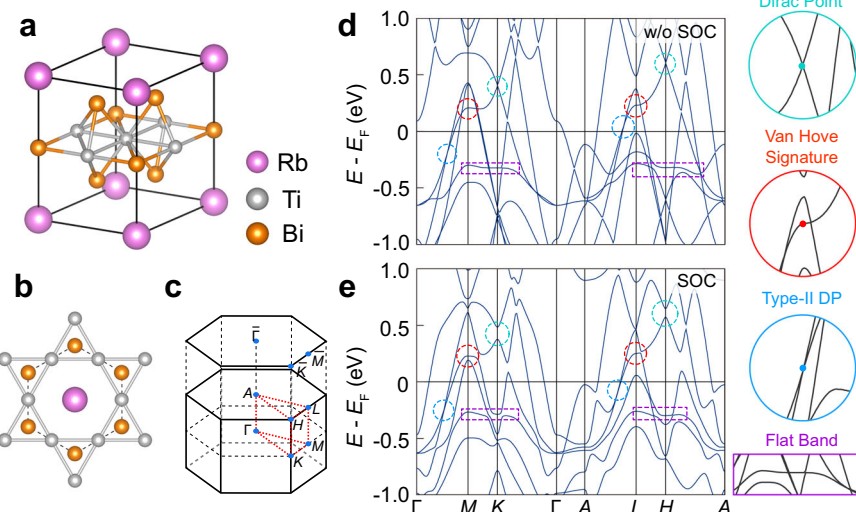

**Fig. 1 | Crystal structure, Brillouin zone and band calculations. a** Crystal structure of $RbTi_3Bi_5$. **b** Top view of the Ti-kagome network. **c** Bulk Brillouin zone (BZ) and projected two-dimensional BZ attached with high-symmetry points are indicated. **d**, **e** Calculated band structure of $RbTi_3Bi_5$ along high-symmetry paths (**d**) without SOC and **e** with SOC.

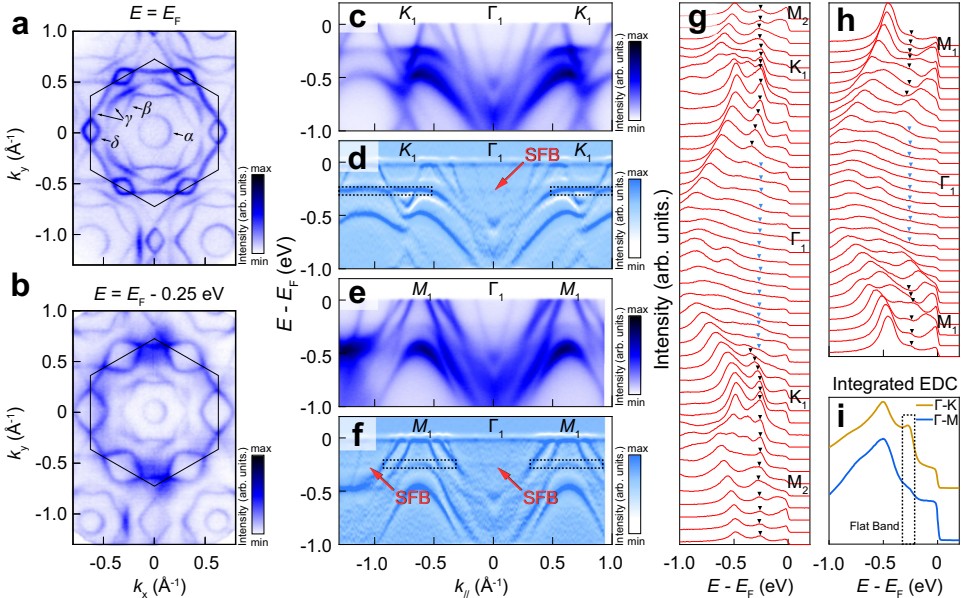

**Fig. 2 | Constant energy contours and flat bands of RbTi$_3$Bi$_5$. a** Fermi surface ($E_F$) and (**b**) constant energy contours at $E_F$ - 0.25 eV of RbTi$_3$Bi$_5$ measured with 66 eV photons. Hexagonal surface Brillouin zones are marked with dark solid lines. **c**, **e** ARPES intensity plot and (**d**, **f**) corresponding curvature derivative plots along $\overline{\Gamma}$-$\overline{K}$-$\overline{M}$ and $\overline{\Gamma}$-$\overline{M}$ high-symmetry direction, respectively. The flat bands are marked with dashed frames and shadow flat bands (SFB) are indicated by red arrows. Panels (**c**–**f**) share the same y-axis. **g** EDCs of **c**. **h** EDCs of (**e**). The flat bands around $K$-$M$ are indicated by black triangles, and the shadow flat determined from the second-order derivation spectrum are indicated by blue triangles. **i** Integrated energy distribution curves along $\Gamma$-$M$ (blue) and $\Gamma$-$K$ (gold) direction. The position of flat band is marked with dark dash box. Panels (**g**–**i**) share the same y-axis.

high density of states (DOS) appears near the $K$ points at around $E_F$ −0.25 eV, and then disappears at higher binding energy [Fig. 2b and Supplementary Fig. 2], indicating the existence of dispersionless bands. To explore the possible dispersionless bands, we present detailed photoemission intensity images and their corresponding second-order derivative plots along the representative $\Gamma$-$K$ and $\Gamma$-$M$ directions, respectively [Fig. 2c–f]. As indicated by red dash frames in Fig. 2c and d, the band structure show almost dispersionless feature at $E_F$ −0.25 eV, which is reminiscent of the characteristic of electronic structure in kagome-lattice. Such flat band feature can be further confirmed by both the non-dispersive peaks in the energy distribution curves (EDCs) [dark triangles in Fig. 2g–f] and shoulders in the integrated EDCs [denoted by red dash line in Fig. 2i] along $\Gamma$ − $K$ and $\Gamma$ − $M$, respectively. Moreover, this flat band at ~ −0.25 eV seems to attach with another "shadow flat band" (SFB) with low intensity, which spreads over the entire BZ (denoted by red arrows in Fig. 2d, f). Further studies on sibling CsTi$_3$Bi$_5$ and KTi$_3$Bi$_5$ have confirmed the presence of SFB features in all members of this family. Although the flat band is missing in regions along $\Gamma$ − $K$ and $\Gamma$ − $M$, as shown in Fig. 1e–f, we cannot simply attribute this to the destructive interference of the electronic wavefunctions present commonly in kagome lattices. However, considering that the binding energy of SFB always coincides with that of the DFT predicted small flat feature around $K$ and $M$, which is at approximately −250 meV in CsTi$_3$Bi$_5$ and RbTi$_3$Bi$_5$, and −300/−580 meV in KTi$_3$Bi$_5$ [Supplementary Fig. 4], the origin of such a SFB in ATi$_3$Bi$_5$ should be closely linked to the small flat band. This phenomenon is comparable to the flat band recently observed near the Fermi surface in CsV$_3$Sb$_5$[21], which also exhibits an additional shadow flat band at the location of the van Hove singularity near the Fermi surface. Additionally, these SFB may also attribute to some rather localized states in ATi$_3$Bi$_5$, e.g., parts of localized titanium 3$d$ electrons due to spin frustration or impurity states with relatively lower intensity and dispersionless feature; Moreover, the $k_z$ broadening of electronic states can also give rise to a "drumhead"-like state at $\Gamma$, more discussion can be found in Supplementary Note 3.

## The type-II Dirac nodal lines in RbTi$_3$Bi$_5$

Distinct from AV$_3$Sb$_5$, Ti-based kagome metals exhibit richer non-trivial band topology in the vicinity of $E_F$. In Fig. 3a and b, we demonstrate the zoomed-in calculated band structure of RbTi$_3$Bi$_5$ around predicted type-II DPs. Along both the $\Gamma - M$ or $A - L$ high-symmetry directions, there exist two linear bands (four when considering the spin degeneracy caused by the combined time-reversal and spatial-inversion symmetries) which cross at a type-II DP at $E_F$-0.2 eV. This DP is exact and protected by both spacial-inversion and time-reversal ($PT$) symmetry when the SOC is ignored, and even the consideration of SOC would just introduce a tiny gap of 9 meV. In total there are six type-II DPs over the $k_x - k_y$ plane, and they all evolve into Dirac nodal lines (DNLs) along $k_z$ since the interlayer couplings are weak. To experimentally verify these type-II DNLs, we performed a detailed photon-energy dependent ARPES measurement, in which all probing cuts were intentionally arranged to fully cover the localization of one DNL in the momentum space, as illustrated in Fig. 3c. Through spectra along these cuts (with different $k_z$) shown in Fig. 3d, we can distinguish a series of type-II Dirac-like band crossings, which are markedly in line with our calculation [red lines in Fig. 3c]. Furthermore, we plot the $k_x$-$k_z$ constant-energy contours integrated within a 50 meV energy window from $E_F$ to $E_F$-0.4 eV across half of the BZ in Fig. 3e. We can discover that the $\beta$ and $\gamma$ bands degenerate to a single curving nodal line exactly at the binding energy of 0.2 eV, as marked by the red arrow. In this way, we can unambiguously confirm the existence of type-II DNLs in RbTi$_3$Bi$_5$. Note that these DNLs are rather close to $E_F$, and they probably make significant topological contribution to transports. By further reducing the valence electrons in RbTi$_3$Bi$_5$, for example, substituting the Ti (3$d^2$) by Sc (3$d^1$) will lead to considerable shift of $d$-states toward the Fermi level. The ability to tune the band features through substitution is of paramount importance as it provides a unique avenue for investigating the direct influence of these features on the material's physical properties. This tunability aspect highlights the rich and diverse nature of the ATi$_3$Bi$_5$ compounds and paves the way for future studies aimed at uncovering the underlying mechanisms governing

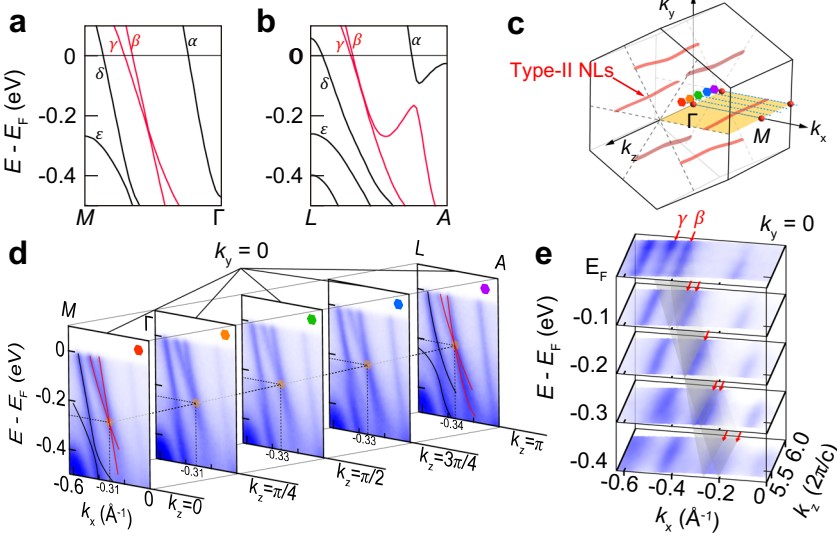

**Fig. 3 | The type-II Dirac nodal lines in RbTi$_3$Bi$_5$. a, b** Calculated bulk bands along $\Gamma$-$M$ and $L$-$A$, respectively. The $\alpha$-$\varepsilon$ mark the first to fifth band from $\Gamma$ to $M$ point. The $\beta$ and $\gamma$ bands forming type-II Dirac cone are highlighted by red lines. **c** Schematic of momentum locations of type-II Dirac nodal lines (red curves, extracted from the calculation in Supplementary Fig. 7) and cut lines along different $k_z$ position (blue dash lines) in the bulk BZ. **d** ARPES intensity plots cut along the $\overline{\Gamma}$-$\overline{M}$ direction at selected $k_z$ positions with photon energies region from 57-66 eV ($k_z = \pi$, $3\pi/4$, $\pi/2$, $\pi/4$ and 0). Orange dots denote the position of type-II Dirac nodes. **e** The stacking constant energy $k_x$-$k_z$ maps along $\overline{\Gamma}$-$\overline{M}$ direction taken at selected energies ($E_F$ to $E_F$-0.4 eV). The red arrows denote the $\gamma$ and $\beta$ bands, the grey planes denote the 3D band dispersion along $k_z$ direction. The c in $k_z$ unit represents the out-of-plane lattice constant.

the observed exotic band features and their connection to intriguing physics.

### The nontrivial topological surface states in RbTi$_3$Bi$_5$

Aside from DNLs, there exists a well-defined topological gap for the $\gamma$ band. Since the $PT$ symmetry is conserved in RbTi$_3$Bi$_5$, a Fu-Kane $\mathbb{Z}_2$ index can be associated to the four time-reversal-invariant points[46]. We calculated their parity eigenvalues [Supplementary Fig. 8], the results are similar to previous calculations[39] [Calculational details can be found in Supplementary Note 6]. We find there are two band inversions which happen around the $\Gamma$ and $K$ points for the $\gamma$ band. Therefore, there should be non-trivial topological surface states (TSSs) in the bulk gap of RbTi$_3$Bi$_5$. We show our calculated and experimental band structures around these TSSs in Fig. 4a, b, respectively. Theoretically, we can indeed observe these TSSs appearing in the bulk band gap in the regions I and II, as shown in Fig. 4a, e, g. Our experimental band dispersion shows overall good agreement with the calculation. In particular, through second derivatives of photoemission intensity images [Fig. 4f, h], all predicted TSSs can be well resolved. We can clearly identify the characteristic split bands around $\overline{K}$ and Rashba-like feature around $\overline{\Gamma}$ except for a small offset and renormalization of binding energy. Furthermore, our photon-energy dependent $k_{//}$-$k_z$ mappings around regions I [Fig. 4c] and II [Fig. 4d] confirm the surface band nature of these resolved bands. Thus, we can verify the existence of non-trivial TSSs in RbTi$_3$Bi$_5$.

### Rotation symmetry breaking in the low-lying electronic structure of CDW-free RbTi$_3$Bi$_5$

One pivotal feature of ATi$_3$Bi$_5$ is the absence of CDW, which is as well regarded as one major advantage over V-based kagome metals in studying the electronic nematicity in kagome lattices[47]. Accordingly, in our photoemission data we did not observe any sign of energy gap or band folding even at the lowest temperature we can achieve [Fig. 5a and Supplementary Fig. 9]. This finding is in sharp contrast to the case of KV$_3$Sb$_5$, in which both the CDW induced energy gap around $M$ and band folding can be well identified, as illustrated in Supplementary Fig. 15. As for V-based kagome metals, the origin of CDW is closely

related to the Fermi surface instability caused by nesting between neighbouring VHSs around $E_F$ and the softening of phonons. However, VHSs in ATi$_3$Bi$_5$ are located well above $E_F$ [right panel of Fig. 5a], and thus the nesting induced Fermi surface instability is absent. It might account for the missing of CDW transition in Ti-based kagome metals. To better understand the formation of CDW in kagome metals, we first estimated the joint DOSs obtained from ARPES autocorrelation (AC-ARPES)[48] determined by Fermi surfaces for both kagome systems [Supplementary Fig. 11]. It describes the phase space ($\boldsymbol{q}$) for electron scatterings from states at $\boldsymbol{k}$ to those at $\boldsymbol{k}+\boldsymbol{q}$. In contrast to KV$_3$Sb$_5$, where the resulting joint DOS exhibits peaks at the CDW wave vectors $\boldsymbol{q}_i$ ($i$=1, 2 and 3)[Supplementary Fig. 11], there exists no similar CDW peaks appearing at the specific wave vector for RbTi$_3$Bi$_5$, as shown in Fig. 5c, suggesting the absence of charge-ordering instabilities therein. Furthermore, we calculated phonon dispersions along high-symmetry paths, as shown in Fig. 5b and Supplementary Fig. 10. Distinct from the case of KV$_3$Sb$_5$, the phonon dispersion of RbTi$_3$Bi$_5$ is free of imaginary phonons and shows rather stable structure, which further eliminates the possibility of CDW formation in this material.

It's worth noting that two recent scanning tunneling microscopy (STM) works reported the electronic nematic phase existing in ATi$_3$Bi$_5$ with the rotation symmetry breaking, which occurs in the absence of the concomitant translation symmetry breaking induced by CDW[47,49]. Although our ARPES results cannot directly reveal the directionality in the low-energy electronic structure at the first sight, the AC-ARPES spectra, which has been applied to give a reasonable count for charge-ordering instabilities of various compounds[48,50–54], indeed exhibits evident anisotropy among the three nominally identical directions. Figure 5c demonstrates the AC-ARPES extracted from the Fermi surface of RbTi$_3$Bi$_5$, in which $\boldsymbol{q}$-vector peaks denoting the scattering channel from $\alpha$ to $\delta$ band just appear at two of three nominally equivalent directions (highlighted by black solid circles), suggesting the breaking of rotational symmetry in the electronic structure. Furthermore, we discovered that this $C_2$ anisotropy persists at 200 K [Supplementary Fig. 14]. Given that the AC-ARPES result based on Fermi surface would accumulate all subtle inhomogeneity in electronic states near $E_F$, such a directionality we discovered might be related to

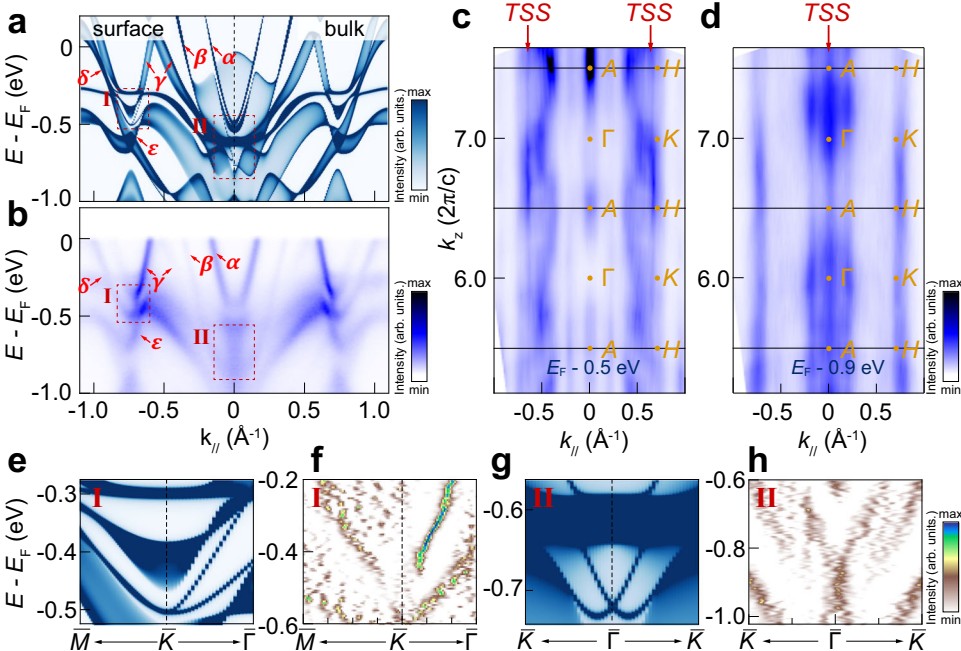

**Fig. 4 | Topological surface states in RbTi₃Bi₅. a** Calculated band structure with (left half) and without surface (right half) states. **b** The $k_{//}$-$E$ ARPES spectra taken along $\overline{K}$-$\overline{\Gamma}$-$\overline{K}$ direction. **c, d** Large-scale constant energy $k_{//}$-$k_z$ map with photon energies ranging from 50-110 eV along $\overline{K}$-$\overline{\Gamma}$-$\overline{K}$ direction taken at $E_F$-0.5 eV and $E_F$-0.9 eV, respectively. Panel (**c, d**) share the same y-axis The solid lines denote the boundary of Brillouin zones and red arrows denote the topological surface states (TSS). The c in $k_z$ unit represents the out-of-plane lattice constant. **e, g** The zoomed-in calculated band structure $\overline{K}$ and $\overline{\Gamma}$ point (red dash boxes I and II in panel **a**). Panel (**e, g**) share the same colorbar with panel (**a**). **f, h** The zoomed-in 2D curvature ARPES spectra taken around $\overline{K}$ and $\overline{\Gamma}$ point (red dash boxes I and II in panel **b**). Panel (**f, h**) share the same colorbar.

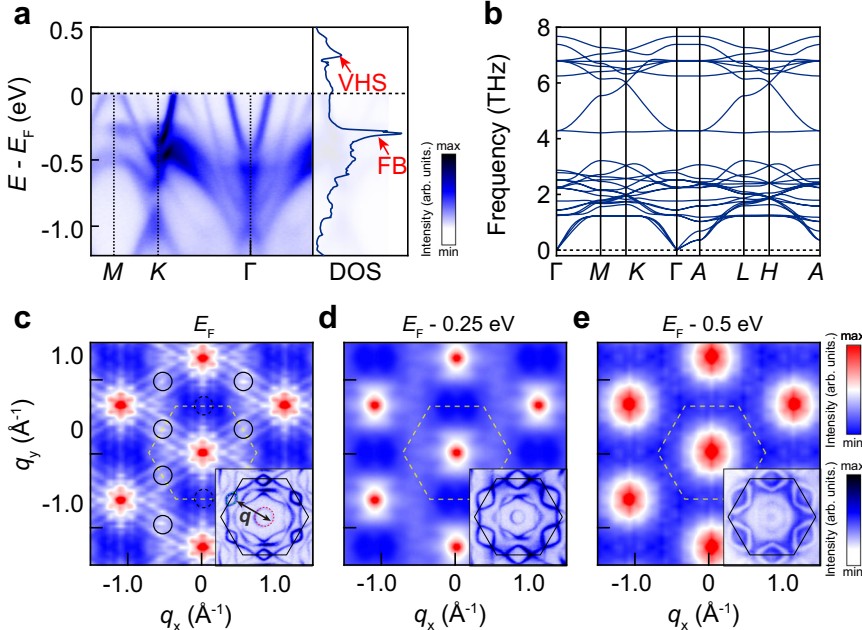

**Fig. 5 | Calculated total density-of-states, phonon spectrum and ARPES auto-correlation. a** ARPES intensity plot of RbTi₃Bi₅ along $M$-$K$-$\Gamma$ and calculated density of states (DOS) are appended on the right side of corresponding panels. The position of van Hove Signature (VHS) and flat band (FB) are marked with red arrows; **b** Calculated phonon spectrum along a high-symmetry path of RbTi₃Bi₅; **c–e** Two-dimensional joint DOS results from experimental Fermi surface of RbTi₃Bi₅ taken at $E_F$, $E_F$ - 0.25 eV, $E_F$ - 0.5 eV. The insets are experimental Fermi surfaces taken at 10 K. The wave vector $q$ is denoted by the black arrow. The main panel of (**c–e**) share the up colorbar on the right side, the inset of panel **c-e** share the bottom colorbar on the right side.

the suggested pure electronic nematic phase in ATi₃Bi₅. We find that the similar feature is also observed for KV₃Sb₅ [Supplementary Fig. 16], which exhibits obvious intensity anisotropy and is consistent with our previous report[55]. Moreover, the detailed evolution of AC-ARPES

results with the binding energy reveals that such $C_2$ anisotropy just appears in the vicinity of $E_F$, and it becomes progressively less obvious with the increasing binding energy and restore to the $C_6$ symmetry [Fig. 5d and Supplementary Fig. 13]. Finally, as the intensity distribution

of Fermi surface map are influenced by the photoemission matrix element effect, we also try to rotate the sample to research how they influence the scattering peaks in autocorrealtion results. In Supplementary Fig. 12, we append all the possible scattering vectors on the Fermi surface of $RbTi_3Bi_5$. Although the scattering intensity of some bands are influenced by the matrix element effect such as the near-hexagon band structures, such as the three parallel lines along $\Gamma - M$ directions and some bright spots around $\Gamma$ points, the intensity peaks associated with scattering between $\alpha$ and $\delta$ bands still rotated with the sample, as shown in Supplementary Fig. 15. This finding implies that such a rotational symmetry breaking would be characteristic of low energy electronic structure of $RbTi_3Bi_5$. We emphasize that the anisotropy in AC-ARPES could be dominated by a variety of extrinsic factors including the renowned matrix element effects, and more conclusive evidences on the nematicity in the electronic structure of $ATi_3Bi_5$ would be still highly desired in the further.

## Discussion

We revealed the coexistence of flat bands, type-II DNLs and $\mathbb{Z}_2$ topological bands in the newly discovered titanium-based kagome metals. We then investigated the underlying mechanism of vanishing CDW states in $ATi_3Bi_5$ by comparing to $KV_3Sb_5$ side-by-side, and confirmed that VHSs around $E_F$ play the key role in forming CDW in kagome metals. We as well discovered that $RbTi_3Bi_5$ shows similar rotation symmetry breaking to $KV_3Sb_5$, implying the possible hidden nematic phase without the intertwined CDW in $ATi_3Bi_5$. Our findings thus provide important insights into non-trivial band topology and rotation symmetry breaking in "135" kagome metals.

*Note added.*

Recently, several photoemission works, which report some results that are similar to part of our findings and were carried out independently by several groups[56–59].

## Methods

### Sample growth and characterization

Single crystals of $RbTi_3Bi_5$ were prepared via the self-flux method. The precursor RbBi was prepared by reacting Rb (purity 99.75%) and Bi granules (purity 99.999%) at 300 °C. Starting materials of RbBi, Bi and Ti powder (purity 99.99%) were mixed in a molar ratio of 3 : 9 : 1, loaded into an alumina crucible, and then was sealed in a quartz ampoule under partial argon atmosphere. The assembly was heated up to 1100 °C in a furnace within 12 h and kept at this temperature for 10 h. It was subsequently slowly cooled down to 800 °C at a temperature decreasing rate of 50 °C/h and kept for 5 h, and then slowly cooled down further to 400 °C at 2 °C/h. Finally, the assembly was taken out from the furnace and decanted with a centrifuge to separate $RbTi_3Bi_5$ single crystal from the flux.

### ARPES experiments

All $RbTi_3Bi_5$ single crystal samples were cleaved in situ at 15 K with a base pressure of better than $6 \times 10^{-11}$ Torr. High-resolution ARPES measurements were performed at the 03U beamline of Shanghai Synchrotron Radiation Facility (SSRF)[60] with liner horizontal polarization lights[61]. In our measurements, light's linear horizontal polarization is parallel to the ground and the incident angle on sample is $\theta = 45°$ with respect to the sample's normal direction, along with the slit direction that is perpendicular to the ground. All data were acquired with a Scienta-Omicron DA30 electron analyzer. The total energy resolution was set to 10 ~ 20 meV depending on the photon energy applied, and the angular resolution was set to be 0.2°. In our experiments, the photon energy region is from 50 to 120 eV, and the inner potential of $RbTi_3Bi_5$ determined by our photon energy dependent measurement is 4 eV. The error may be introduced by the resolution of the beamline and analyzer.

### Band calculations

First-principles calculations based on density functional theory (DFT) were performed using the Vienna ab initio simulation package (VASP) which adopts the projector-augmented wave method[62]. The energy cutoff was set at 400 eV and exchange-correlation functional of the Perdew-Burke-Ernzerhof (PBE) type[63] is used for both the structural relaxations and electronic structures calculations. The convergence criteria for the total energy and forces are set to $10^{-6}$ eV and 0.001 eV/Å, respectively. The Wannier tight-binding models are built through the *wannier90* code[64]. The surface states are calculated with iterative Green function method[65]. Phonon spectrums are calculated with *phonopy* code[66], and density functional perturbation method is used as implemented in VASP. The parity eigenvalues of the high-symmetry points are analyzed using the *irvsp* code[67]. We have tested different calculational schemes and parameters to avoid possible numerical error.

## Data availability

The authors declare that the main data supporting the findings of this study are available within the paper and its Supplementary Material. Extra data are available from the corresponding authors upon request.

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

## Acknowledgements

We acknowledge the National Natural Science Foundation of China (Grants No. U2032208, 92065201, 12222413, 12004405 and 12174257), the support by the National Key R & D program of China (Grant No. 2022YFB3608000 and 2020YFA0309601), the Shanghai Science and Technology Innovation Action Plan (Grant No. 21JC1402000), the Natural Science Foundation of Shanghai (Grant No. 22ZR1473300, 23ZR1482200). Y.F.G. was sponsored by Double First-Class Initiative Fund of ShanghaiTech University. Part of this research used Beamline 03U of the Shanghai Synchrotron Radiation Facility, which is supported by $ME^2$ project under Contract No.11227902 from National Natural Science Foundation of China. The authors also thank the support from Analytical Instrumentation Center (#SPST-AIC10112914).

## Author contributions

Z.C.J., Z.T.L., and D.W.S. performed the ARPES experiment and analyzed the resulting data. H.Y.M. and J.P.L. performed the theoretical calculations. W.X., X.Q.L., and Y.F.G. synthesized and characterized the single crystals. Z.T.L., J.P.L., Y.F.G., and D.W.S. supervised the project. Z.C.J., Z.T.L., Z.H.L., J.S.L., S.C., Y.C.Y., J.Y.D., J.Y.L., Z.H., Y.X.Q., J.J.S., W.C.J. contributed to the development and maintenance of the ARPES systems, beamline and related software development. Z.C.J., Z.T.L., and D.W.S. wrote the manuscript with input from all coauthors.

## Competing interests

The authors declare no competing interests.
