## [Peer Review File · Nature Communications]

REVIEWER COMMENTS

Reviewer #1 (Remarks to the Author):

This manuscript describes a first-principle and ARPES study of RbTi₃Bi₅, a kagome compound recently synthesized. It is isostructural to the celebrated AV₃Sb₅ family, but with a different effective doping. Therefore, an ARPES investigation is indeed very interesting. The DFT calculation is similar to earlier reports (ref. 39-40 for example).

The ARPES data presented here are of high quality and they arguably bring information useful for the community. They are compared to DFT calculations to « confirm » the presence of some features, such as flat bands or topological surface states. This is a standard way to proceed, but I find that one does not learn very much on the physics of this material in the process. Especially, there are a few points that require more discussions.

1- The « flat band » the authors refer to is only « flat » in a small window around KM in the calculation (and in the map of Fig. 2b). In contrast, the authors observe a « flat band » extending everywhere. In Fig.2g, they call the part around Gamma « shadow flat band » but this term is not justified anywhere in the text. The authors should comment on the origin of this feature. Finally, the authors argue that it is close to EF, but -0.25eV is still rather far compared to all temperature scales considered here. The authors should explain why they think it can be important.

2- Similarly, the two types of Dirac points are far from EF and unlikely to play a direct role. As a demonstration of non-trivial topology, the reasoning of ref. 39 is reproduced in SI. Ref. 40 claims that this reasoning cannot be applied here, which is not mentioned. Some more comments about this would be welcome.

3- In abstract, it is written that « We found VHS play an important role in the formation of CDW in kagome metals ». I suppose they mean that the lack of CDW could be explained by the shifting of the VHS above EF, as predicted in the calculation. As this is just a speculation based on the calculation, I would not call this a « finding ».

4- As recognized by the authors themselves, ARPES matrix element effects should be seriously considered before claiming (or even suggesting) nematicity from the autocorrelation plots of Fig. 5. On the contrary, no details are given on the experimental setup. The methods section mentions horizontal polarization, in which direction ? Does the pattern rotate with the sample ? This seems a basic test to perform before reporting such data.

The interest of the compound and the data quality justify attention to this manuscript, but my recommendation to publish it or not in Nature Communication would strongly depend on the answer the authors can give to the above points.

Reviewer #2 (Remarks to the Author):

The authors present ARPES studies of Kagome material RbTi₃Bi₅. This work is in a highly topical area and will clearly be of interest to those working on this category of materials. The study follows several very interesting papers on the AV₃Sb₅ material system, exhibiting numerous interesting phenomena such as superconductivity, non-trivial topology, charge density waves, and nematicity.

This work should, after revision, find a place in a widely read high-quality journal. However, I am not convinced of this work's suitability for Nature Communications. I feel that the manuscript, while nice in several ways, lacks new physical insights and that message overall is that RbTi₃Bi₅ is similar to other Kagome materials. A lot of what is discussed is very closely linked to discussions of flat bands and topology that have been discussed in this material family for a couple of years. A more interesting point relates to that of CDW/nematicity. But even there papers such as [47] and [49] have appeared discussing the lack of a CDW and the appearance of nematicity in Ti-based Kagome materials. I think the manuscript would be suitable for npjQM or PRL, but that the work falls short of what's needed for

Nature Communications.

1. My main technical complaint relates to the flat band physics. While I can see that there is a certain flatish character to the measured electronic structure, the text seems quite exaggerated.
 - a. I think the data contradict the claim of a flat band across the entire BZ as stated in the text. The features marked with the black triangles in Fig 2 show appreciable dispersion and if I track the feature from K1 (or M1) towards gamma it seems to disperse quite a lot even if the black arrows are omitted.
 - b. The exact meaning of "shadow band" should be clarified. If the meaning is just "low intensity", I would suggest using this more straightforward terminology.
 - c. What is the criterion for placing the blue triangles on Figures 2g&h? I cannot see any feature in many of these EDCs.
2. The text might need to be updated to account for arXiv: 2212.07958. This preprint seems to have appeared on Dec 15. I'm told to disregard preprints that appears *after* Dec 15, so this seems to be a case where this preprint should be considered? This is obviously a borderline case, and a time when input from the editor would be helpful, but I think it would be better to avoid claiming precedence in the abstract "For the first time, we experimentally demonstrate the coexistence of flat bands and multiple non-trivial topological states, including type-II Dirac nodal lines and non-trivial Z2 topological surface states therein." Given the STM preprint arXiv:2211.16477 argues for nematicity in CsTi3Bi5, the "for the first time" claim is already something that is arguable.
3. The manuscript requires an additional careful check as there are typos such as "Van Hove signatrue" in Fig. 1 among a few others.

Reviewer #3 (Remarks to the Author):

In this manuscript, Jiang et al. report the first angle-resolved photoemission spectroscopy (ARPES) measurements of a kagome metal RbTi3Bi5, material isostructural to the heavily investigated superconductors AV3Sb5 (A = Rb, K, Cs). Chemical substitution in the kagome sublattice leads to significantly reduced band filling, shifting the positions of various singularities and topologically non-trivial bands in the electronic structure with respect to the Fermi level. As a result, flat bands and topological surface states (TSSs) are brought into the vicinity of the chemical potential, while the van Hove singularities (VHSs), thought to be responsible for exotic charge order in the vanadium-based Kagome metals, are placed in the unoccupied part of the band structure. The authors extensively characterize the interesting electronic states by means of comparative ARPES for ATi3Bi5, and AV3Sb5 compounds, further complemented by electronic band structure and phonon calculations. The data is discussed in the context of the absence of charge density wave (CDW) order in this sub-class of Kagome materials as well as the emergence of electronic nematicity. The study addresses possible origins of the exotic quantum states in a hot topic class of compounds, and will be attractive to a broad readership of Nature Communications. Before I can recommend this manuscript for publication, I would like to pose following questions and comments:

- The introduction promotes the idea that the hole doping in ATi3Bi5 compounds brings the flat and topologically non-trivial bands sufficiently close to the Fermi level to overcome the barrier to triggering exotic ordered phases. However, the states of interest in the Ti-based compounds are not as close to the chemical potential as the VHSs are in AV3Sb5 – in fact, the calculation in Fig. 1 locates the two types of singularities at a similar distance to EF (-250 meV vs +200 meV). This is in apparent contradiction with the claim that "VHSs in ATi3Bi5 are located well above EF". If 200 meV is large enough an energy scale to prevent the CDW formation order, why should it be small enough to produce other phenomena?
- Why did the authors decide to focus on Rb-containing compound as the main compound for the investigation, as opposed to the other members of the ATi3Bi5 series? Could they comment further on the more prominent flat band structure in KTi3Bi5?
- What is the orbital character of the highlighted bands?
- In the discussion of autocorrelation ARPES for KV3Sb5, I do not see intensity anisotropy but rather different shapes of the features (dogbone vs cigar). How to understand such a result?
- The photon energy range and assumed inner potential should be given for the kz-dependent scans in Figs. 3(d)-(e), 4(c)-(d).
- The manuscript would benefit from more careful proof-reading. In its present shape, it contains

some typos and inconsistencies (e.g. "[Figs. 2c-]" on page 3; "wace vector" on page 5; 200 K given in the caption to Supp. Fig. 10 while the rest of the text refers to 120 K), and some of the sentences are a bit difficult to follow.

Reviewer #1 (Remarks to the Author):

This manuscript describes a first-principle and ARPES study of RbTi₃Bi₅, a kagome compound recently synthesized. It is isostructural to the celebrated AV₃Sb₅ family, but with a different effective doping. Therefore, an ARPES investigation is indeed very interesting. The DFT calculation is similar to earlier reports (ref. 39-40 for example).

The ARPES data presented here are of high quality and they arguably bring information useful for the community. They are compared to DFT calculations to confirm the presence of some features, such as flat bands or topological surface states. This is a standard way to proceed, but I find that one does not learn very much on the physics of this material in the process. Especially, there are a few points that require more discussions.

Our response:

We gratefully thank Referee #1 for his/her review and praise for our data quality. Also, we fully understand his/her concerns. In the following paragraphs, we will try to address all his/her comments point by point, and we hope that our response and revisions to the manuscript could satisfy him/her.

Our work is not only important to the field working in this class of materials, but will also draw significant interest from the community of strongly correlated physics. The unique features of kagome band structures, including Dirac cones, van Hove signatures, and flat bands, make the kagome lattice system a perfect platform to study the interplay between strong correlation effects and nontrivial band topology. Thus, quantum materials with kagome lattices continue to captivate researchers in the field of condensed matter physics, especially in the community of strongly correlated physics. The "135"-type vanadium-based Kagome material is a typical example to demonstrate the rich correlated states that could emerge from kagome systems, such as unconventional charge orders, electronic nematicity, and superconductivity. In our manuscript, we report the discovery of one another comparative "135"-type titanium-based kagome material that exhibits a nematic electronic state even in the absence of charge density waves. Our work implies that the nematic phase may be a universal phenomenon for correlated metals with kagome structures. In this sense, our work is of great value in understanding the universal properties of the correlated states in kagome metals.

1- The « flat band » the authors refer to is only « flat » in a small window around KM in the calculation (and in the map of Fig. 2b). In contrast, the authors observe a « flat band » extending

everywhere. In Fig.2g, they call the part around Gamma « shadow flat band » but this term is not justified anywhere in the text. The authors should comment on the origin of this feature. Finally, the authors argue that it is close to EF, but -0.25eV is still rather far compared to all temperature scales considered here. The authors should explain why they think it can be important.

Our response:

We would like to express our gratitude to Referee #1 for highlighting the issue of confusion caused by our statements. We apologize for any misunderstandings that may have arisen due to these misleading statements. We intentionally used the term "shadow flat band" to refer to the unexpected weak flat band feature, indicated by yellow dashed frames in Figs. R1(a-c), that appears around Γ in ATi_3Bi_5 . This feature is seemingly superimposed onto the common flat band caused by the destructive interference of electronic wavefunctions in kagome lattices. Our research on sibling compounds CsTi_3Bi_5 and KTi_3Bi_5 has revealed that these shadow flat bands are present in all members of this family. Our finding was also supported by several other works on the electronic structure of ATi_3Bi_5 [Yang *et al.*, arXiv:2212.04447; Hu *et al.*, arXiv:2212.07958 (2022)].

Since this shadow flat band is a characteristic feature that spreads over the entire Brillouin zone, it cannot be reproduced by DFT calculations. Therefore, we cannot simply attribute it to the destructive interference of electronic wavefunctions commonly found in kagome lattices. However, as the binding energy of the shadow flat bands coincides with that of the DFT-predicted small flat feature around K and M (-250 meV in CsTi_3Bi_5 and RbTi_3Bi_5 and -300/-580 meV in KTi_3Bi_5), we assume that **the origin of such a shadow flat band in ATi_3Bi_5 is closely linked to the small flat band**. We propose several possible origins for this shadow flat band. Firstly, it might originate from rather localized states in ATi_3Bi_5 , such as localized vanadium $3d$ electrons due to spin frustration or impurity states with relatively lower intensity and dispersionless features. Secondly, this feature may be related to the k_z broadening of electronic states at Γ . While the in-plane electron hopping is restricted in kagome lattices, interlayer electron hopping remains significant in real three-dimensional stacked kagome lattices. Consequently, dispersive bands develop along the k_z direction. Our DFT calculations have indicated that the band structure around Γ is largely dispersive along k_z , resulting in a band structure evolution from the electron-like to "drumhead"-like, as shown by the highlighted area around Γ/A in Fig. R2(b). As our ARPES measurements were mainly performed using VUV photons, the mean escape lengths of the resulting photoelectrons are limited. Therefore, the photoemission spectra may be more influenced by the k_z broadening effect. Nevertheless, we note that more further researches are strongly desired to pinpoint the definitive origin of this exotic shadow flat band.

Fig. R1: Comparison of band structure and corresponding second-order derivation spectrum along $K\text{-}\Gamma\text{-}K$ of RbTi_3Bi_5 , CsTi_3Bi_5 and KTi_3Bi_5 . The yellow frames highlight the position of flat bands and yellow arrows in (d)-(e) highlight the position of flat features with low intensity.

Regarding to the importance of flat band discovered in ATi_3Bi_5 , we firstly note that flat bands in most kagome lattices are not located at Fermi level, e.g., $\text{CoSn} \sim -0.27$ eV [*Nat. Commun.* 11, 4004 (2020)], $\text{Fe}_3\text{Sn}_2 \sim -0.2$ eV [*Phys. Rev. Lett.* 121, 096401 (2018)], and $\text{YMn}_6\text{Sn}_6 \sim -0.5$ eV [*Nat. Commun.* 12, 3129 (2021)]. As for AV_3Sb_5 , the predicted flat band is located at $E_F - 1.2$ eV, which is so deep in binding energy. To our best knowledge, there exists just one previous paper reporting the not well-defined flat band in CsV_3Sb_5 [Hu *et al.*, *Sci Bull* 67, 495 (2022)], and thus we had rather limited knowledge on flat bands, one of most characteristic features of kagome lattices, in AV_3Sb_5 . Comparatively, the flat feature is much closer to the Fermi level and more well-defined in Ti-based 135 family, and the study on flat bands in ATi_3Bi_5 would thus be helpful in better understanding of AV_3Sb_5 .

Additionally, in the Ti-based 135 family, the flat band in RbTi_3Bi_5 is closer to the Fermi level than that of KTi_3Bi_5 , distinct from the V-based 135 family, suggesting the immense potential in the tunability of binding energy of flat bands in these 135 series materials. We are expecting a further tuning of flat bands towards the Fermi level in ATi_3Bi_5 by substituting more kinds of elements, such as doping Sc on the Ti site or Sn on the Bi site, which would introduce more hole carriers. Also, we note that the flat band and type-II Dirac nodal lines are approximately located at the same binding energy and momentum position in ATi_3Bi_5 , as shown in Fig. R3. This serendipitous crossing would lead to tons of interesting phenomena that deserve further researches [e.g., *Nature* 584, 59 (2020); *PRX* 11, 031017 (2021); *PRB* 104, L081104 (2021); *Phys. Rev. B* 96, 155137 (2017); *npj Quantum Mater.* 7, 73 (2022); *Advances in Physics: X*, 3, 1473052 (2018); *J. Phys.: Condens. Matter* 32, 165501 (2020), etc.]

For the above reasons, nearly all recent ARPES reports on Ti-based 135 family have

spent much of their testimony putting great emphasis on the discovery of flat bands therein, which as well proves the recognition of this importance of flat bands in ATi_3Bi_5 from a side.

In the revised manuscript, we have redefined the “shadow flat bands” in the main text. Besides, we have added the corresponding discussion about its possible origin and significance of flat bands in ATi_3Bi_5 into Supplementary Note 3.

Fig. R2 (a) The second-order derivation ARPES spectrum cut along the M - K - Γ - K - M direction; (b) The DFT calculated band structure of RbTi_3Bi_5 , the blue lines calculated along the high-symmetry line M - K - Γ - K - M direction and the yellow lines calculated along the high-symmetry line L - H - A - H - L .

Fig. R3 Second-order derivation spectrum and enlarged region along M - Γ - M direction of RbTi_3Bi_5 .

2- Similarly, the two types of Dirac points are far from EF and unlikely to play a direct role. As a demonstration of non-trivial topology, the reasoning of ref. 39 is reproduced in SI. Ref. 40 claims that this reasoning cannot be applied here, which is not mentioned. Some more comments about this would be welcome.

Our response:

We would like to extend our gratitude to Referee #1 for this valuable suggestion. While it is true that the two types of Dirac points in RbTi_3Bi_5 series materials are not in close proximity to the Fermi level, the coexistence of multiple topologies and flat bands in a single material is still compelling [e.g., *Phys. Rev. Lett.* 106, 236803 (2011); *Phys. Rev. X* 4, 011010 (2014); *Nat. Mater.* 19, 163 (2020); *Nat Commun.* 12, 2732 (2021)]. This implies that the electronic properties of RbTi_3Bi_5 are highly tunable and provide a fertile ground for emerging phenomena such as

superconductivity and nematic states. Notably, we observed that the flat bands and the type-II Dirac nodal lines are located at extremely similar binding energies [Fig. R3], indicating a possible interplay between band topology and electronic correlations in RbTi₃Bi₅. In this way, interesting physics such as CDW, SDW, superconducting, and topological charge orders might occur upon further tuning. Thus, this discovery might provide useful information for the community in understanding correlations and topology phenomena in kagome materials.

As mentioned by the referee, Ref. 40 claimed that the \mathbb{Z}_2 topological invariant cannot be calculated using the parity products at the TRIM points due to the absence of the continuous band gap. However, this is not the case if we apply the theory of symmetry indicators [*Nature* 566, 486 (2019); *J. Phys.: Condens. Matter* 32, 263001 (2020)]. According to these theories, parity eigenvalues at the TRIM points can be used to diagnose the band topology regardless whether there is a continuous band gap or not. For ATi₃Bi₅, which has both time reversal and space inversion symmetry, a more general topological invariant is

$$\chi_{BS} = \mathbb{Z}_2^3 \times \mathbb{Z}_4$$

where the 3 \mathbb{Z}_2 indices correspond to the 3 weak indices of Fu-Kane criterion and the strong \mathbb{Z}_2 index is promoted to a \mathbb{Z}_4 factor. We can still use the \mathbb{Z}_2 index if we only would like to learn whether the compound is topological non-trivial or not. However, to identify whether the compound is a “insulator” or semimetal, we agree that a further analysis of the band crossings is necessary.

In the revised manuscript, we have added this discussion into Supplementary Note 6.

3- In abstract, it is written that « We found VHS play an important role in the formation of CDW in kagome metals ». I suppose they mean that the lack of CDW could be explained by the shifting of the VHS above EF, as predicted in the calculation. As this is just a speculation based on the calculation, I would not call this a « finding ».

Our response:

We appreciate Referee#1 for pointing out this mistake. We fully agree with him/her on this comment and recognize that the term is misleading. According to our photoemission spectral result, we could just determine that the VHS in ATi₃Bi₅ is well above the Fermi level, and thus we deduced that the VHS in ATi₃Bi₅ should not play an dominating role in driving the CDW transition as in AV₃Sb₅. This is NOT a finding but just a speculation. In our revised manuscript, we have altered our statements accordingly.

4- As recognized by the authors themselves, ARPES matrix element effects should be seriously considered before claiming (or even suggesting) nematicity from the autocorrelation plots of Fig. 5.

On the contrary, no details are given on the experimental setup. The methods section mentions horizontal polarization, in which direction? Does the pattern rotate with the sample? This seems a basic test to perform before reporting such data.

Our response:

We appreciate Referee#1 for bringing to our attention the omission of some critical experimental details in our manuscript. We understand that this might cause inconvenience for our readers. Thus, we have included an illustration of our experimental setup in Fig. R4, which highlights the light's linear horizontal polarization A_p that is parallel to the ground and the $\theta = 45^\circ$ angle at which it is incident on the sample with respect to the sample's normal direction, along with the slit direction that is perpendicular to the ground.

Considering that the ARPES intensity $I_{photoelectron} \propto \langle f | \mathbf{A} \cdot \hat{\mathbf{p}} | i \rangle$, an arbitrary electron passing through the slit is mirror symmetric with the M_x mirror plane, as shown in Fig. R4(a), which can be simplified by the mirror operator $\hat{\pi}_y | y \rangle = -| y \rangle$. The incident light polarization A could be divided into two components: (1) The in-plane component $A_y = A_p \cdot \cos\theta$, which gives $\hat{\pi}_y (A_y \cdot \hat{p}_y) \hat{\pi}_y^\dagger = -A_y \cdot \hat{p}_y$ and $\pi_{y,A_y} = -1$; (2) The out-of-plane component $A_z = A_p \cdot \sin\theta$, which gives $\hat{\pi}_y (A_z \cdot \hat{p}_z) \hat{\pi}_y^\dagger = A_z \cdot \hat{p}_z$ and $\pi_{y,A_z} = +1$. For the final state we have $\hat{\pi}_y | f \rangle \equiv | f \rangle$, thus $\pi_{y,f} \equiv +1$. Given that the d -orbitals of Ti atoms and p -orbitals of Bi mainly contribute to the bands near the Fermi level, we analyzed the matrix element effect of the incident light polarization under this setup. We concluded that the matrix element effect on the d -orbital bands is summarized in Table R1, from which we deduce that all d - and p -orbitals should be observed.

Furthermore, we attempted to measure rotated samples according to Referee#1's suggestion, as shown in Fig. R5. When the sample was rotated by -15° and -30° , both corresponding autocorrelation results rotate along with the sample. **In particular, the characteristic feature of anisotropy in the amplitudes between the three nominally identical directions keeps rotating by the same degrees**, as shown in Figs. R5(e-f). **This result unambiguously indicates that the matrix element effects should not dominate the anisotropy in the autocorrelation.** Following Referee#1's suggestion, we have found that these supplementary data further strengthen our original argument of the intrinsic nematicity (not like that discovered in AV_3Sb_5 related to the three-dimensional CDW) in the electronic structure of ATi_3Bi_5 . We hope that these details would provide greater clarity and transparency regarding our experimental procedures and results, and we would like to thank Referee#1 once again for his/her valuable suggestion to help further reinforce our results.

In the revised manuscript, we revised the ARPES experiment configuration in Method part. We also modified the description of nematic phase in the main text and added the new experiment results to Supplementary Figure 14 and Supplementary Note 5.

Fig. R4 (a) The experimental configuration in our experiment. The A_p represent the direction of linear horizontal polarization, the green plane represents photoemission plane parallel to analyzer slit direction. (b) The Fermi surface mapping of RbTi_3Bi_5 taken at 66 eV and 5 K. (c) The sketch of orbital distribution at Fermi surface.

Orbitals	d_{xz}	d_{yz}	d_{xy}	$d_{x^2-y^2}$	d_{z^2}	p_x	p_y	p_z
$\pi_{y,i}$	-1	+1	-1	+1	+1	-1	+1	+1
$\pi_{y,f}$	+1	+1	+1	+1	+1	+1	+1	+1
A_y	-1	-1	-1	-1	-1	-1	-1	-1
$I_{\text{photoelectron}}, A_y$	zero	Non-zero	zero	Non-zero	Non-zero	zero	Non-zero	Non-zero
A_z	+1	+1	+1	+1	+1	+1	+1	+1
$I_{\text{photoelectron}}, A_z$	Non-zero	Non-zero	Non-zero	Non-zero	Non-zero	Non-zero	Non-zero	Non-zero
Total	Non-zero	Non-zero	Non-zero	Non-zero	Non-zero	Non-zero	Non-zero	Non-zero

Tab. R1 The analysis of the spectrum intensity $I_{\text{photoelectron}}$ of orbitals considering photoemission matrix element effect under our measurement setup.

Fig. R5 (a)-(c) ARPES maps of Fermi surface with sample were set to (a) 0° , (b) -15° respect to (a), and (c) -30° respect to (a); (d)-(f) The autocorrelation results of (a)-(c), respectively.

Reviewer #2 (Remarks to the Author):

The authors present ARPES studies of Kagome material RbTi₃Bi₅. This work is in a highly topical area and will clearly be of interest to those working on this category of materials. The study follows several very interesting papers on the AV₃Sb₅ material system, exhibiting numerous interesting phenomena such as superconductivity, non-trivial topology, charge density waves, and nematicity. This work should, after revision, find a place in a widely read high-quality journal. However, I am not convinced of this work's suitability for Nature Communications. I feel that the manuscript, while nice in several ways, lacks new physical insights and that message overall is that RbTi₃Bi₅ is similar to other Kagome materials. A lot of what is discussed is very closely linked to discussions of flat bands and topology that have been discussed in this material family for a couple of years. A more interesting point relates to that of CDW/nematicity. But even there papers such as [47] and [49] have appeared discussing the lack of a CDW and the appearance of nematicity in Ti-based Kagome materials. I think the manuscript would be suitable for npjQM or PRL, but that the work falls short of what's needed for Nature Communications.

Our response:

We appreciate Referee#2 for his/her comments about the topicality of our research, and we as well expect that this work would be of great interest to those working on this category of materials. While, we don't agree with Referee#2 that our manuscript lacks new physical insights and that message overall is that RbTi₃Bi₅ is similar to other kagome materials. Next, we would like to summarize the importance of our findings in ATi₃Bi₅.

Firstly, we experimentally reported the unconventional flat bands in ATi₃Bi₅. To our best knowledge, our manuscript, posted as the preprint arXiv: 2212.02399 on Dec. 5, 2022, is the first work experimentally reporting the flat band in ATi₃Bi₅. On one hand, we found that, the flat band feature in ATi₃Bi₅ is spreading over the whole BZ, which cannot be reproduced by DFT calculations, distinct from flat bands confined to finite momentum space commonly discovered in other previously reported kagome materials. Notably, this finding has been confirmed by several separate groups [Yang *et al.*, arXiv:2212.04447 (2022); Liu *et al.*, arXiv:2212.04460, (2022); Wang *et al.*, Chinese Phys Lett, 40, 037102 (2023); Hu *et al.*, arXiv:2212.07958 (2022); Zhou *et al.*, arXiv:2301.01633 (2022)]. Thus, we could not simply attribute this unique feature to the destructive interference of electronic wavefunctions in kagome lattices. On the other hand, although there still exists an energy dispersion of around 80 meV, this flat band feature discovered in ATi₃Bi₅ is of the smallest energy dispersion among all previously reported flat bands in kagome compounds, namely of the most ideal flat band character. Although to date there have been no definitive origin of this

unique flat band feature, further investigations on that in ATi_3Bi_5 would shed light on the understanding of flat-band related electronic correlations in kagome lattices.

Secondly, we identified multiple topological non-trivial states in ATi_3Bi_5 , which have not been observed or even predicted in kagome lattices. Previous works on ATi_3Bi_5 have predicted the existence of non-trivial Z_2 topological surface states therein but did not present any experimental confirmation. Our work provided the first spectroscopic evidence on this topological surface state. Besides, in our manuscript, we as well unveiled the other one non-trivial band topology, i.e., the type-II Dirac nodal lines, which were firstly discovered in kagome lattices to our best knowledge. We note that the coexistence of multiple band topologies and flat bands in one single material is rather compelling, which has led to various novel physics [e.g., *Phys. Rev. Lett.* 106, 236803 (2011); *Phys. Rev. X* 4, 011010 (2014); *Nat. Mater.* 19, 163–169 (2020); *Nat Commun.* 12, 2732 (2021)]. In particular, the discovered flat bands and type-II Dirac nodal lines are located at extremely similar binding energies, rendering ATi_3Bi_5 unique as a kagome structure for exploring the interplay between correlations and nontrivial band topology. Our finding would provide useful information for the community in understanding correlations and topology phenomena in kagome materials.

Lastly, our work implies that the nematic phase might be a universal phenomenon for correlated metals with kagome structures. We agree with Referee#2 that papers such as [47] and [49] have appeared discussing the lack of a CDW and the appearance of nematicity in Ti-based kagome materials. However, we would like to highlight the advantage and uniqueness of ARPES evidence on the electronic nematicity in solids. In previous STM measurements, the electronic nematicity or rotational symmetry breaking was determined by discovering different amplitude of CDW peaks in Fourier transform of topography. Usually, the electronic nematicity is mainly discussed in the momentum space. Although the real and momentum representations can be related by a Fourier transform, the philosophical implications derived from these two techniques (STM vs. ARPES) are very different. STM mainly focuses on localized electronic states, which is easily influenced by surface defects and impurity (for example, the previous controversy in STM results of chiral and C_2 rotation symmetry in KV_3Sb_5 finally turned out to be a defect related phenomenon [Jiang *et al.*, *Nat. Mater.* 20, 1353 (2021); Kang *et al.*, *Nat. Phys.* 18, 301 (2022); Shumiya *et al.*, *PRB* 104, 035131 (2021)]). Comparably, ARPES can directly obtain the overall electronic states driven by periodic lattice structure and not so sensitive to the defects. Moreover, ARPES can measure the electrons with mean free path length of ~ 10 Å, which is more “bulk” sensitive than STM. Therefore, in other systems exhibiting electronic nematicity, e.g., unconventional superconductors, ARPES studies have always provided the most straightforward and conclusive clues to the electronic nematic phase. As for ATi_3Bi_5 , our study has offered the first spectroscopic evidence and comprehensive understanding of the electronic nematicity from the view of reciprocal space. Thus, we believe that

our ARPES finding of ATi_3Bi_5 is a significant advance in this field.

We have fully understood all questions and concerns raised by Referee#2. In the following paragraphs and revised manuscript, we have tried our best to address all those comments and concerns raised by Referee#2 point by point. We hope that our responses and the revision to the manuscript could satisfy him/her.

1. My main technical complaint relates to the flat band physics. While I can see that there is a certain flatish character to the measured electronic structure, the text seems quite exaggerated. a. I think the data contradict the claim of a flat band across the entire BZ as stated in the text. The features marked with the black triangles in Fig 2 show appreciable dispersion and if I track the feature from K_1 (or M_1) towards Γ it seems to disperse quite a lot even if the black arrows are omitted.

Our response:

We are grateful to Referee #2 for this feedback to our manuscript regarding the flat band physics. We firstly would like to apologize for the black arrows in Fig. 2 which might cover up some raw data and thus cause unnecessary misleadingness. In fact, we fully agree with Referee #2 that there exists energy dispersion for the band feature from K_1/M_1 to Γ , and it is not strictly flat. According to our second-order derivation photoemission intensity plot [Fig. R6] and corresponding energy distribution curves [Fig. R7], we can deduce that the energy dispersion of this flat band around K - M is around 80 meV. Note that this dispersion in ATi_3Bi_5 is much smaller than the flat band observed in AV_3Sb_5 [around 220 meV, Hu *et al.*, *Sci. Bull.* 67, 495 (2022)]. Actually, this is the smallest energy dispersion among all previously reported flat bands in kagome compounds, including the well-known flat band in CoSn [~120 meV, Liu *et al.*, *Nat. Commun.* 11(1) (2022)], FeSn [~160 meV, Kang *et al.*, *Nat. Mater.* 19, (2020)], Fe_3Sn_2 [~170 meV, Lin *et al.*, *Phys. Rev. Lett.*, 121, 096401 (2018)], YMn_6Sn_6 [~150 meV, Li *et al.*, *Nat. Commun.*, 12(1) (2021)]. We conjectured that this flat band extending to the Γ point in ATi_3Bi_5 might be largely influenced by both the not perfect Ti-kagome lattice with Bi inserted in the center and sizable spin-orbital coupling, which thus result in the observable energy dispersion of flat band.

Besides, we have found that this flat band connects with another extremely flat feature with the low intensity toward Γ , which has never been reported in other kagome materials. Such an extra shadow feature around Γ is even flatter, with a nearly zero energy dispersion according to our second-derivative plots [Fig. R6]. We found that the combination of both flat band feature is of the characteristic feature of spreading over the whole BZ, **which was also confirmed by some other recent ARPES works on ATi_3Bi_5** [Yang *et al.*, *arXiv:2212.04447*; Hao *et al.*, *Chinese Phys Lett.*, 40, 037102 (2023)].

In our revised manuscript, we have revised the text to clarify that these flat bands in ATi_3Bi_5 are dispersive and their flatness is not perfect. We have also updated the figure caption to reflect this observation accordingly.

Fig. R6 (a) Second-order derivation spectrum along $M\text{-}\Gamma\text{-}M$ direction of RbTi_3Bi_5 ; (b) Second-order derivation spectrum along $M\text{-}K\text{-}\Gamma\text{-}K$ direction of RbTi_3Bi_5 . The yellow arrows denote the shadow flat bands.

Fig. R7 Energy distribution curves (EDCs) along $M\text{-}K\text{-}\Gamma$ direction of RbTi_3Bi_5 . The EDCs around $M\text{-}K$ direction are highlighted by blue curves, and the peak position of flat band are marked by red short line.

b. The exact meaning of “shadow band” should be clarified. If the meaning is just “low intensity”, I would suggest using this more straightforward terminology.

Our response:

Here, we intentionally used the “shadow flat band” to specifically refer to the unexpected weak flat band feature appearing around Γ , as highlighted by yellow dashed frames in Figs. R1(a-c). We found that the binding energy of this shadow flat bands always coincides with that of DFT predicted small flat feature around K and M , namely -250 meV in CsTi_3Bi_5 and RbTi_3Bi_5 and $-300/-580$ meV in KTi_3Bi_5 , just as the shadow of real flat bands caused by the destructive interference of the electronic wavefunctions. Thus, we prefer to name it as the shadow flat band.

In the revised manuscript, we have redefined the “shadow flat bands” in the main text. Besides, we have added the corresponding discussion about the possible origin of this shadow flat into Supplementary Note 3.

c. What is the criterion for placing the blue triangles on Figures 2g&h? I cannot see any feature in

many of these EDCs.

Our response:

We deeply apologize for any confusion due to the positioning of blue triangles in Figs. 2(g) and 2(h). These triangles were placed to mark the dispersion of the low-intensity band, which is approximately one-tenth of the intensity of the common flat band feature along K - M . For energy distribution curves covering a large area, the intensity of the flat feature is often reduced by the adjacent band structure. Therefore, its accurate identification is always challenging. To address this issue, we first applied the second derivative plots to determine the position of the shadow flat band around Γ , as presented in Figs. 2(d) and 2(f) in the main text, and Fig. R6. Then we directly appended all this data points onto EDCs for a guide of eyes. In the revised manuscript, we have clarified that in the corresponding caption.

*2. The text might need to be updated to account for arXiv: 2212.07958. This preprint seems to have appeared on Dec 15. I'm told to disregard preprints that appears *after* Dec 15, so this seems to be a case where this preprint should be considered? This is obviously a borderline case, and a time when input from the editor would be helpful, but I think it would be better to avoid claiming precedence in the abstract "For the first time, we experimentally demonstrate the coexistence of flat bands and multiple non-trivial topological states, including type-II Dirac nodal lines and non-trivial Z_2 topological surface states therein." Given the STM preprint arXiv:2211.16477 argues for nematicity in $CsTi_3Bi_5$, the "for the first time" claim is already something that is arguable.*

Our response:

We appreciate Referee#2 for reminding us the nice manuscript arXiv: 2212.07958. Our manuscript, which was posted as the preprint arXiv: 2212.02399 on Dec. 5, 2022, to our best knowledge, is the first ARPES works on ATi_3Bi_5 . Thus, we were unable to know the preprint arXiv: 2212.07958 when our work was submitted. While, we agree with Referee #2 and we have updated our work accordingly and cited these similar works including arXiv: 2212.07958 in the revised manuscript.

In addition, Referee#2's suggestion to avoid claiming precedence in the abstract regarding the coexistence of flat bands and multiple non-trivial topological states is reasonable. We acknowledge that our claim "for the first time" is indeed arguable. We have revised our main text including the abstract to reflect this and to avoid making any claims that might be refuted by future research.

3. The manuscript requires an additional careful check as there are typos such as "Van Hove signatrue" in Fig. 1 among a few others.

Our response:

We really apologize for these typos in our manuscript caused by our carelessness. In the revised manuscript, we have tried our best to correct all such errors.

Once again, we appreciate Referee#2's careful reading of our manuscript and valuable feedback. We hope that our revised manuscript would address his/her concerns and right now meet the standard of Nature communications.

Reviewer #3 (Remarks to the Author):

In this manuscript, Jiang et al. report the first angle-resolved photoemission spectroscopy (ARPES) measurements of a kagome metal RbTi₃Bi₅, material isostructural to the heavily investigated superconductors AV₃Sb₅ (A = Rb, K, Cs). Chemical substitution in the kagome sublattice leads to significantly reduced band filling, shifting the positions of various singularities and topologically non-trivial bands in the electronic structure with respect to the Fermi level. As a result, flat bands and topological surface states (TSSs) are brought into the vicinity of the chemical potential, while the van Hove singularities (VHSs), thought to be responsible for exotic charge order in the vanadium-based Kagome metals, are placed in the unoccupied part of the band structure. The authors extensively characterize the interesting electronic states by means of comparative ARPES for ATi₃Bi₅, and AV₃Sb₅ compounds, further complemented by electronic band structure and phonon calculations. The data is discussed in the context of the absence of charge density wave (CDW) order in this sub-class of Kagome materials as well as the emergence of electronic nematicity. The study addresses possible origins of the exotic quantum states in a hot topic class of compounds, and will be attractive to a broad readership of Nature Communications. Before I can recommend this manuscript for publication, I would like to pose following questions and comments:

Our response:

We are grateful to Referee #3 for helping review our manuscript and his/her positive feedback to our study of the kagome metal RbTi₃Bi₅. Moreover, we fully understand his/her questions and concerns. In the following paragraphs, we will try to address all his/her comments point by point, and we hope that our response and revisions to the manuscript could satisfy him/her.

- The introduction promotes the idea that the hole doping in ATi₃Bi₅ compounds brings the flat and topologically non-trivial bands sufficiently close to the Fermi level to overcome the barrier to triggering exotic ordered phases. However, the states of interest in the Ti-based compounds are not as close to the chemical potential as the VHSs are in AV₃Sb₅ – in fact, the calculation in Fig. 1 locates the two types of singularities at a similar distance to EF (-250 meV vs +200 meV). This is in apparent contradiction with the claim that “VHSs in ATi₃Bi₅ are located well above EF”. If 200 meV is large enough an energy scale to prevent the CDW formation order, why should it be small enough to produce other phenomena?

Our response:

We would like to thank Referee#3 for his/her insightful comment on our introduction. We as

well realized that -250 meV is not small enough to affect the transport properties. However, in the Ti-based 135 family, we have noticed that the flat band in RbTi₃Bi₅ is closer in binding energy to the Fermi level than that of KTi₃Bi₅, distinct from the V-based 135 family, strongly suggesting the immense potential in the tunability of binding energy of flat bands in these 135 series materials. We are thus expecting a further tuning of flat bands towards the Fermi level in ATi₃Bi₅ by substituting more kinds of elements, such as doping Sc on the Ti site or Pb/Sn on the Bi site. Moreover, we have noticed that many works also focus on the exotic flat bands and topology electronic structure located at $E_F - 250$ meV in ATi₃Bi₅ series materials [Yang *et al.*, arXiv:2212.04447 (2022); Liu *et al.*, arXiv:2212.04460 (2022); Wang *et al.*, *Chinese Phys. Lett.* 40, 037102 (2023); Hu *et al.*, arXiv:2212.07958 (2022); Zhou *et al.*, arXiv:2301.01633 (2022)], which as well proves the importance of flat bands and topological non-trivial states discovered close to the Fermi level from a side.

In the revised manuscript, we have modified all these statements correspondingly.

- Why did the authors decide to focus on Rb-containing compound as the main compound for the investigation, as opposed to the other members of the ATi₃Bi₅ series? Could they comment further on the more prominent flat band structure in KTi₃Bi₅?

Our response:

We are grateful to Referee #3 for his/her valuable question. In this work, we have investigated all three members of this Ti-based 135 family (CsTi₃Bi₅, RbTi₃Bi₅, and KTi₃Bi₅) simultaneously, as shown in Supplementary Fig. 3. General speaking, we found that all of three compounds exhibit rather similar characteristics, namely the coexistence of flat bands and multiple non-trivial topological states, including type-II Dirac nodal lines and non-trivial Z₂ topological surface states. Just in some minor aspects, they demonstrate some tiny differences. Therefore, we picked RbTi₃Bi₅ out as the typical example, simply for its relatively superior photoemission spectral quality to the other two, which we hope that would allow for more reliable band structure analysis including small band splitting.

Regarding flat bands in KTi₃Bi₅, we found them to be slightly distinct from those in RbTi₃Bi₅ and CsTi₃Bi₅. In KTi₃Bi₅, we observed two flat bands located at different binding energies along K-M direction, while in RbTi₃Bi₅ and CsTi₃Bi₅ we just observed one. As shown in Fig. R8(b) and (c), the constant energy surface taken at $E_F - 0.3$ eV and $E_F - 0.58$ eV of KTi₃Bi₅ both show high density of states around the edge of Brillouin zones. These flat bands can also be clearly observed along the high symmetric K-M-Γ-K-M direction; we marked these two flat bands with red arrows in Fig. R8(e). Besides, we note that when we alter the photon energy to the L-H-A-H-L direction, the flat feature

at $E_F - 0.58$ eV becomes much clearer. Additionally, both flat bands extend a feature with low intensity to the Γ point, which is not visible in our calculations. The flat feature with low intensity at around $E_F - 0.3$ eV can be visible in the raw spectrum [Fig. R8(d)] and second-derivative spectrum [Fig. R8(e)], respectively. Given that the binding energy of this low intensity flat bands always coincides with that of DFT predicted small flat feature around K and M, namely -0.25 eV in CsTi₃Bi₅ and RbTi₃Bi₅ and -0.30/-0.58 eV in KTi₃Bi₅, the low intensity flat features across the entire Brillouin zone should tightly linked with the flat bands confined in K - M direction, but the detailed origin of multiple flat bands in KTi₃Bi₅ is still waiting for further investigation.

In the revised manuscript, we have added this part into Supplementary Materials Fig. S5 and Supplementary Note 5.

Fig. R8 (a)-(c) The constant energy maps taken at (a) E_F ; (b) $E_F - 0.3$ eV; (c) $E_F - 0.58$ eV of KTi₃Bi₅, respectively. (d) The band structure along M - K - Γ - K - M measured at 74 eV; (e) The second-order derivation spectrum of (d); (f) The band structure along L - H - A - H - L measured at 86 eV; (g) The second-order derivation spectrum of (f).

- *What is the orbital character of the highlighted bands?*

Our response:

The orbital character of the highlighted bands is presented in the Fig. R9. The Fermi surface are mainly constituted by d -orbitals of Ti-atoms and p -orbitals of Bi atom, the calculated orbital distribution at Fermi surface is presented in Fig. R10(b), and more detailed orbital calculations are

presented in Figs. R10(d)-(e). According to the calculation, we can summarize the orbital characters of different bands in Fig. R10(c), where the inner circular pocket (red) around Γ point mainly originates from the Bi p_z orbital, the hexagon-like one (green) is from Ti d_{xz}/d_{yz} orbitals, the another hexagon-like pocket (yellow) is mainly from Ti $d_{xy}/d_{x^2-y^2}$ orbitals, and the rhombic-like and triangle-like bands (gray) are mainly attributed to Ti p_x/p_y orbitals.

In the revised manuscript, we have added this part in Supplementary Fig. S3 and Supplementary Note 2.

Fig. R9 (a) The ARPES map at Fermi surface of RbTi₃Bi₅; (b) The calculated distribution of orbitals on the Fermi surface of RbTi₃Bi₅; (c) The sketch plot of orbital characterization of RbTi₃Bi₅; (d) The calculated d-orbital distribution along high-symmetry cut line; (e) The calculated p-orbital distribution along high symmetry cut line.

- In the discussion of autocorrelation ARPES for KV₃Sb₅, I do not see intensity anisotropy but rather different shapes of the features (dogbone vs cigar). How to understand such a result?

Our response:

We fully understand the referee's concern. We would like to provide a clearer explanation on this point.

Our previous ARPES study has reported the breaking of C_6 rotation symmetry in KV₃Sb₅, as illustrated in Fig. R10(a). The CDW folding bands in KV₃Sb₅ have been reported to be

largely reconstructed by the nematic charge order, and the folding bands only form along the uniaxial direction [Jiang *et al.*, *arXiv:220801499*, 2022]. In Fig. R10(a), we have marked the different scattering channels by double-head arrows along different Γ - M directions: q_1 represents the scattering from the circular-like band around Γ to the folded circular-like band around M , q_2 represents that from the circular-like band around Γ to the “ ∞ ”-like band around M . The autocorrelation calculation enables us to transfer the k -space spectrum into q -space, as demonstrated in Fig. R10(b). Since the C_6 rotation symmetry is broken in KV_3Sb_5 , the band structure of KV_3Sb_5 also exhibits an anisotropic feature.

To highlight the anisotropic feature in the q -space image of autocorrelation, we directly compared the three intensity curves taken along Cut-1, Cut-2 and Cut-3, as shown in Fig. R10(c). We observed that the intensity at q_2 along all three directions are approximately equal, while the adjacent q_1 peak along Cut-3 is much higher than those along Cut-1 and Cut-2. Moreover, the anisotropy feature can be more clearly seen from the autocorrelation result of the constant energy intensity map at $E_F - 0.2$ eV, as illustrated in Fig. R10(d)-(f). We note that the intensity of q_2 peaks is nearly equal for all three directions. However, the q_1 peak is rather outstanding along Cut-3 while it is evidently suppressed along both Cut-1 and Cut-2. Both bulges at q_1 taken along Cut-3 for E_F and $E_F - 0.2$ eV reflect relatively stronger correlation between the pristine and CDW folded bands along q_1 for KV_3Sb_5 , unambiguously indicative of the breaking of original C_6 symmetry therein.

In the revised manuscript, we have added this data analysis in Supplementary Note 5.

Fig. R10 (a) The ARPES map of KV_3Sb_5 at Fermi surface and 14 K; (b) The autocorrelation results calculated from (a); (c) The intensity distribution curves along cut 1, cut 2 and cut 3 in (b); (d) The ARPES map of KV_3Sb_5 at $E_F - 0.2$ eV and 14 K; (e) The autocorrelation results calculated from (d); (f) The intensity distribution curves along cut 1, cut 2 and cut 3 in (e).

- The photon energy range and assumed inner potential should be given for the k_z -dependent scans in Figs. 3(d)-(e), 4(c)-(d).

Our response:

We apologize for the missing of important parameters in the main text. According to the photon energy dependent ARPES measurement, we have determined that the inner potential of RbTi₃Bi₅ is about 4 eV, which is consistent with several recent works [Hu *et al.*, *arXiv:2212.07958* (2022); Zhou *et al.*, *arXiv:2301.01633* (2023)]. In Figs. 3(d)-(e), the photon energies range from 57 ~ 66 eV which have covered more than half of BZ along the k_z direction, and in Figs. 4(c)-(d), the photon energies range from 50 ~ 110 eV. In the revised manuscript, we have indicated all these parameters in the corresponding captions.

- The manuscript would benefit from more careful proof-reading. In its present shape, it contains some typos and inconsistencies (e.g. “[Figs. 2c-]” on page 3; “wace vector” on page 5; 200 K given in the caption to Supp. Fig. 10 while the rest of the text refers to 120 K), and some of the sentences are a bit difficult to follow.

Our response:

We apologize for any confusion caused by the typos and inconsistencies in our manuscript. In the revised manuscript, we have carefully proofread the manuscript to try to correct all these issues and ensure that the text is clear and consistent throughout. We appreciate again for Referee#3’s time and feedback in reviewing our work.

REVIEWER COMMENTS

Reviewer #1 (Remarks to the Author):

The authors have clarified some points of the manuscript about the shadow flat band or the calculation of topological properties, which I appreciate. However, much of their answer does not really address precisely the questions. For example, I was not confused by the shadow flat band, I just found that it has to be defined and discussed in the main text, this was a clear lack. Moreover, it's not because others have observed similar features or that flat bands in other kagome compounds are even further from EF that these remarks are not relevant.

But my main interest is about the nematic state deduced from autocorrelation functions of ARPES spectra. Nematicity is such a subtle state that, different from Referee 2, I find it important to discuss its possible occurrence with different techniques. Moreover ARPES can probe a much higher temperature range than STM. Out of interest for this measurement, I would really like the authors to go further into this analysis.

- Could they be more specific about the scattering probably giving rise to the 3 parallel lines along GM? In the text, they mention scattering between alpha and delta, but it's not clear to me that it should give rise to 3 parallel lines. On the other hand, I would really expect a strong feature from the nearly hexagonal gamma shape, which does not seem to be present. Or maybe there is enough curvature in gamma to favor nesting directions slightly away from GM, hence the three lines? Surely, they could comment on this with small models of their nice FS.

- There is so little distortion in the apparent Fermi Surface symmetry, that the experiment as a function of sample rotation (Fig. S14) appears very crucial to me to give ground to this claim. I thank the authors for adding this. With bear eyes, the main asymmetry of the FS is the higher intensity of the feature along the vertical GK direction in (a), certainly due to polarization effects, as it is weakest in the horizontal GK direction in (c). It turns out that the intensity of the AC function is smallest in (d) and largest in (f). While the reason for such a correlation would not be obvious to me, I think a more quantitative analysis would help to clarify the origin and reliability of the effect.

- I feel there is a contradiction between the sentence « This result unambiguously indicates that the matrix element effects should not dominate the anisotropy in the autocorrelation. » in the authors' reply to my comment and the sentence « However, we emphasize that the anisotropy in AC-ARPES could be affected by a variety of extrinsic factors including the renowned matrix element effects... » at the end of the manuscript. Which is the one the authors believe?

I would not recommend publication of this work if the meaning of this observation is not clarified.

Reviewer #2 (Remarks to the Author):

I thank the authors for their reply. As I said from the outset the manuscript is highly topical, of overall good quality, and valuable, and should be published in a widely read high-quality journal. In my previous report, I did raise a criticism that the work is closely linked to prior studies of other Kagomé materials of the general form $RbTi_3Bi_5$, but with different atoms on the Rb and Ti sites. I appreciate the authors' reply pointing to many ways that their manuscript differs from prior studies and I completely agree that there are new interesting results here. I still feel that a stronger reply would have pointed to a clear conceptual difference from prior works or have provided a more compelling argument that the differences observed are of major importance. Having said this, I would tend to agree with the overall sense of the reply that the manuscript will attract substantial attention within the very active Kagomé materials area. My suggestion to the editor would, therefore, be to publish the manuscript in Nature Communications, although this is not a case where I feel strongly one way or the other.

Regarding my numbered points:

1. This issue is mostly solved, but it looks like Figure 2d,f uses the acronym "FB" in red where they actually mean the "shadow flat band". This holds a bit more potential for confusion for readers who are not reading all the text in detail since the end of the caption says that flat bands are marked by red dashed lines. I would suggest circling the flat band in black panels d,f,&i to match the color of the arrows in panel g (of course, there is no need to use black, but it would be nice to match the colors). I would then use a colored arrow or box to mark the shadow flat band (SFB) in panels d&f and match

the color to that used in the arrows of g (green may work well).
The other issues are adequately addressed.

Reviewer #3 (Remarks to the Author):

The authors have addressed my comments and made substantial revisions to the text which I believe have improved the clarity of the motivation and strength of the presented arguments, especially concerning the nematicity.

As mentioned previously, the data is of very high quality. It is also fair to say that this material system already attracts a large interest in the community as evidenced by the arXiv papers which were not available at the time of original submission but are now acknowledged by the authors. With this in mind, I think a publication in Nature Communications may be warranted, however, I have two outstanding comments/suggestions:

- A common criticism of the manuscript among all the reviewers was the lack of direct influence of the observed features on the interesting physics such as superconductivity in this material. I think the authors' response to this is reasonable but is not very well conveyed in the revised text. For example, the comment about the expected tunability of the band structure by substitution should be emphasized more.

- I appreciate the distinction that the authors make between the "flat bands" well-visible in the raw ARPES spectra and the "shadow flat bands" which appear rather as an intensity cut-off in the raw spectra. I would like to see this distinction carried over to the figures where the shadow features are simply labelled "FB".

List of Changes:

1. On page 3 Fig. 2, we circle the flat bands with dark dash frames and mark those shadow flat bands with green arrows, we use abbreviation “SFB” to refer to those shadow flat bands in Fig. 2d and 2f. In Fig. 2g and 2h, we use green triangles to refer to those energy cut-offs of shadow flat bands;
2. On page 3 Fig. 2, we inserted “The flat bands are marked indicated by green arrows” in the middle of caption of Fig. 2;
3. On page 3 line 9, we inserted the “Additionally, we speculate that these some rather localized states more discussion can be found in Supplementary Note 3.” behind the sentence “This phenomenon is comparable to near the Fermi surface.”;
4. On page 5 line 18, we inserted the “By further reducing the valence electrons in RbTi₃Bi₅ their connection to intriguing physics.” behind the sentence “Note that these DNLs make significant topological contribution to transports.”;
5. On page 5 line 8, “Supplementary Fig. 14” is replaced by “Supplementary Fig. 15”;
6. On page 5 line 10 from the bottom, “Supplementary Fig. 13” is replaced by “Supplementary Fig. 14”;
7. On page 5 line 5 from the bottom, “Supplementary Fig. 15” is replaced by “Supplementary Fig. 16”;
8. On page 6 line 3, “Supplementary Fig. 12” is replaced by “Supplementary Fig. 13”;
9. On page 6 line 3, we delete the sentence “However, we emphasize that the anisotropy in AC-ARPES could be affected by a variety of extrinsic factors including the renowned matrix element effects, and more conclusive evidences on the nematicity in the electronic structure of ATi₃Bi₅ would be still highly desired in the further” and replace them with following more accurate statements;
10. On page 6 line 3, we inserted “Finally, as the intensity distribution of Fermi surface map are influenced by the photoemission matrix element effect, and more conclusive evidenced on the nematicity in the electronic structure of ATi₃Bi₅ would be still highly desired in the further.” behind the sentence “ Moreover, the detailed evolution of AC-ARPES results increasing binding energy and restore to the C₆ symmetry.”;
11. On Supplementary Information page 9, we inserted a new Supplementary Figure 12 to describe the scattering vectors on the Fermi surface of RbTi₃Bi₅;
12. On Supplementary Information page 11, we modified the Supplementary Figure 15 (original Fig. 14);
13. On Supplementary Information page 14 line 10, we inserted a paragraph “ In Supplementary

Fig. 12, we identified the scattering vectors three parallel lines along Γ -M directions, as shown in Supplementary Fig. 12f.”;

14. On Supplementary Information page 14 line 21, “Supplementary Fig. 13” is replaced by “Supplementary Fig. 14”;

15. On Supplementary Information page 14 line 10, we inserted a paragraph “Meanwhile, we also attempted to measure rotated samples, the extract origin of the rotation symmetry breaking, which still wait for further exploration.”;

16. On Supplementary Information page 14 line 37, “Supplementary Fig. 15(a)” is replaced by “Supplementary Fig. 16(a)”;

17. On Supplementary Information page 14 line 41, “Supplementary Fig. 15(b)” is replaced by “Supplementary Fig. 16(b)”;

18. On Supplementary Information page 15 line 3, “Supplementary Fig. 15(c)” is replaced by “Supplementary Fig. 16(c)”;

19. On Supplementary Information page 16 line 6, “Supplementary Fig. 14(d)-(f)” is replaced by “Supplementary Fig. 15(d)-(f)”.

Note: All the variation has been highlighted in our manuscript by using the red text.

REVIEWER COMMENTS

Reviewer #1 (Remarks to the Author):

The authors have clarified some points of the manuscript about the shadow flat band or the calculation of topological properties, which I appreciate. However, much of their answer does not really address precisely the questions. For example, I was not confused by the shadow flat band, I just found that it has to be defined and discussed in the main text, this was a clear lack. Moreover, it's not because others have observed similar features or that flat bands in other kagome compounds are even further from EF that these remarks are not to relevant.

Our Response:

We sincerely appreciate Reviewer#1's feedback on the revised version of our manuscript, and we apologize for any misunderstanding of Reviewer #1's questions in our previous response. We agree that the shadow flat band should be clearly defined and discussed in the main text of the manuscript. In the revised version, we have reformulated the definition of the shadow flat band and included some discussion about it in the main text. A more detailed discussion on this issue has been included in the Supplementary Information (SI) Note. 3.

But my main interest is about the nematic state deduced from autocorrelation functions of ARPES spectra. Nematicity is such a subtle state that, different from Referee 2, I find it important to discuss its possible occurrence with different techniques. Moreover, ARPES can probe a much higher temperature range than STM. Out of interest for this measurement, I would really like the authors to go further into this analysis.

Our Response:

We sincerely appreciate the encouragement and valuable suggestions provided by Reviewer #1 regarding the utilization of ARPES to investigate the nematicity. In the following paragraph, we would like to fully address all Reviewer#1's new concerns point to point.

- Could they be more specific about the scattering probably giving rise to the 3 parallel lines along GM? In the text, they mention scattering between alpha and delta, but it's not clear to me that it should give rise to 3 parallel lines.

Our Response:

Fig. R1 (a) Fermi surface mapping of RbTi₃Bi₅; (b) Autocorrelation calculation of Fermi surface; (c) Extracted Fermi surface model with triangle- and rhombic-like bands of RbTi₃Bi₅; (d) Simulated Autocorrelation result according to the model in (c).

The presence of three parallel lines is mainly attributed to the scattering vector among those triangle- and rhombic-like band structures around the edge of Brillouin zone. The scattering between states in one parallel edge along the Γ - M direction contributes to the middle one among the three parallel lines. Additionally, the parallel edges between triangles located at adjacent K points contributes to lines on either side among the three parallel lines, as highlighted by the pink dashed lines. We indicated these scattering vectors with a series of pink dashed arrows (q_4), which correspond well to the positions of parallel lines in Fig. R1(b). Furthermore, we reconstructed a Fermi surface model along the Brillouin zone boundary, **consisting only of triangular- and rhombic-pockets**, as depicted in Fig. R1(c). Utilizing this model, we performed the simulation of autocorrelation and indeed obtained parallel lines along the Γ - M direction [Fig. R1(d)], which rather resemble the AC-ARPES results shown in Fig. R1(b). This simulation strongly suggests that the nesting between states on these triangular- and rhombic-pockets should dominate the three parallel lines shown in the AC-ARPES. **We also note that the parallel lines in the joint-DOS have as well been observed in STM measurements of AV₃Sb₅ [Li et al., Nat. Phys. 19, 637-643 (2023); Wu et al. Nat. Phys. 10.1038/s41567-023-02031-5], and the authors also attributed these parallel lines to scattering between triangle-like bands around K points.**

Besides, we would like to emphasize that the scattering vectors from α to δ bands (q_3) would

contribute to the formation of bright spots near M point, as highlighted by dark dashed circles in Fig. R1(b). However, such a scattering would not give rise to the three parallel lines as mentioned by Reviewer #1.

In the revised manuscript, we have added the discussion about the parallel lines in the AC-ARPES in the Supplementary Note. 5.

On the other hand, I would really expect a strong feature from the nearly hexagonal gamma shape, which does not seem to be present. Or maybe there is enough curvature in gamma to favor nesting directions slightly away from GM, hence the three lines? Surely, they could comment on this with small models of their nice FS.

Our Response:

We sincerely appreciate Reviewer#1's constructive comments, and his/her observation regarding the absence of a strong feature from the nearly hexagonal gamma shape is really insightful to us.

In Fig. R2(a), we identified all the scattering vectors induced by the nearly hexagonal band structures, including q_1, q_1', q_1'' and q_2, q_2' , and we have incorporated them into the displayed AC-ARPES results in Fig. R2(b). The scattering vectors denoted by the red and blue arrows give rise to a series of bright spots and lines around both Γ and K points, but they are obscured by some other stronger features. To clarify this result, we constructed a small Fermi surface model based on the ARPES results and performed simulations to obtain the corresponding autocorrelation, as shown in Figs. R2(c-f). Figure R2(c) displays the scattering vectors q_1, q_1' and q_1'' generated by the parallel edges of the large hexagon surrounding Γ point and the parallel edges from adjacent Brillouin zones, respectively. The simulation result demonstrates that these vectors primarily give rise to a bright line surrounding the Γ point [Figs. R2(d)]. On the other hand, the scattering vector originating from the small hexagon mainly produces three bright spots near the K point, as indicated by q_2 and q_2' in the Fig. R2(f). Here, the scattering between states from the small hexagon would also contribute a little to the middle line among the three parallel lines mentioned above. However, we note that it is the nesting between states on triangular- and rhombic-pockets that dominates the three parallel lines shown in the AC-ARPES.

In the revised manuscript, we have added Fig. R1 and R2 to Supplementary Fig. 12. And we have included this part discussion into Supplementary Note. 5.

Fig. R2 (a) Fermi surface model extracted from the experiment Fermi surface map; (b) Autocorrelation calculation based on (a); (c) Extracted Fermi surface model of larger hexagon bands; (d) Autocorrelation simulation based on (c); (e) Extracted Fermi surface model of smaller hexagon bands; (f) Autocorrelation simulation based on (e).

- There is so little distortion in the apparent Fermi Surface symmetry, that the experiment as a function of sample rotation (Fig. S14) appears very crucial to me to give ground to this claim. I thank the authors for adding this. With bear eyes, the main asymmetry of the FS is the higher intensity of the feature along the vertical GK direction in (a), certainly due to polarization effects, as it is weakest in the horizontal GK direction in (c). It turns out that the intensity of the AC function is smallest in (d) and largest in (f). While the reason for such a correlation would not be obvious to me, I think a more quantitative analysis would help to clarify the origin and reliability of the effect.

Our Response:

We are grateful to Reviewer#1 for his/her insightful suggestion. As pointed out by him/her, the

distortion of Fermi surface symmetry in RbTi_3Bi_5 is minimal, making it challenging to clearly distinguish these differences. Besides, previous QPI reports suggested that the six-fold rotation symmetry breaking in ATi_3Bi_5 likely arises from anisotropic electronic scattering between the central band around the Γ point (α -band) and the bands near the edges of the Brillouin zone (δ -bands) [Ref. 47, 49]. The magnitude of this effect in the spectra is so small that quantitative measurement becomes challenging. Therefore, we employ the AC-ARPES method to capture these subtle changes.

Despite the polarization effect causing intensity differences between the Γ - M and Γ - K directions, leading to background anisotropy in the autocorrelation maps as pointed out by the reviewer, our experimental emphasis always lies on the intensity anisotropy derived from the scattering between α and δ pockets. These anisotropies are observed as bright spots near the M points of the Brillouin zone in the spectra, as evidenced by the model and autocorrelation simulation depicted in Figs. R3(a) and R3(b). To be more quantitative, we extracted the intensity distribution curves along three different Γ - M directions as shown in Fig. R3(e). We could observe a noticeable intensity peak at the position indicated by q_3 in two out of the three directions (yellow and blue dashed cut lines). While the spectra weight on the three parallel lines appeared to be fixed during the sample rotation, influenced by the polarization effect, the intensity peaks associated with q_3 still rotated with the sample. These q_3 peaks were always distinguishable from the yellow and blue cuts but indistinguishable from the red one in -15° and -30° rotated samples, as depicted in Fig. R3(h) and R3(k). **This result unambiguously indicates that the matrix element effects should not dominate the anisotropy of q_3 vectors in the autocorrelation.** We speculate that this may be attributed to a small distortion of the δ -bands, which have been primarily contributed by Bi $p_{x,y}$ orbitals and coupling orbitals of Bi $p_{x,y}$ and Ti d_{xy} orbitals in a previous literature [Ref. 49]; on the other hand, the sibling $\text{Cs}(\text{Ti}_x\text{V}_{1-x})_3\text{Sb}_5$ was recent reported to processes anisotropic electron-phonon coupling kink on the Brillouin zone edges near Fermi surface, which can also introduce a small anisotropic band distortion [Wu et al., Nat. Phys. 10.1038/s41567-023-02031-5], and we suspect that a similar phenomenon might also occur in ATi_3Bi_5 . Thus, these bands might undergo deformation in nematic phases. However, due to the complexity of Fermi surface scattering and the small magnitude of the distortion, we were unable to fully pin down the exact origin of the rotation symmetry breaking right now.

In the revised manuscript, we have modified the Supplementary Fig. 15 and include this part discussion in Supplementary Note. 5

Fig. R3 (a) Fermi surface model with α - and δ -bands of RbTi₃Bi₅; (b) Autocorrelation results calculated from Fermi surface model in (a); (c, f, i) ARPES intensity maps taken at Fermi surface with Γ - M direction (c) 0° , (f) -15° and (i) -30° respect to the analyze slit direction; (d, g, k) AC-ARPES spectra calculated from Fig. R3(c), (f) and (i) respectively; (e, h, k) The intensity distribution curves along three nonequivalent Γ - M directions in Fig. R4(d), (g) and (j), their directions are marked by corresponding colored dash line in Fig. R4(d), (e) and (j).

- I feel there is a contradiction between the sentence « This result unambiguously indicates that the matrix element effects should not dominate the anisotropy in the autocorrelation. » in the authors' reply to my comment and the sentence « However, we emphasize that the anisotropy in AC-ARPES could be affected by a variety of extrinsic factors including the renowned matrix element effects... » at the end of the manuscript. Which is the one the authors believe?

Our Response:

We apologize for any confusion arising from the discrepancy between our previous response

and main text. In our previous responses, we intended to highlight the significance of the matrix element effect and discuss its impact on the experimental results. While it did result in certain changes in the intensity of the spectra (e.g., the intensity of the three parallel lines), we observed that its influence on the symmetry-breaking feature was relatively weak. Meanwhile, in our previous response, we have tried to eliminate the influence of the matrix element effect by rotating the sample. Also, we apologize for unintentionally neglecting the necessary modifications to the main text in our previous response. In this response, we have eliminated the sentence "However, we emphasize that the anisotropy in AC-ARPES could be affected by a variety of extrinsic factors including the renowned matrix element effects..." from the main text and included a discussion on the matrix element effect in the concluding paragraph. Please see the detailed list of changes in 9-10.

Reviewer #2 (Remarks to the Author):

I thank the authors for their reply. As I said from the outset the manuscript is highly topical, of overall good quality, and valuable, and should be published in a widely read high-quality journal. In my previous report, I did raise a criticism that the work is closely linked to prior studies of other Kagomé materials of the general form RbTi_3Bi_5 , but with different atoms on the Rb and Ti sites. I appreciate the authors' reply pointing to many ways that their manuscript differs from prior studies and I completely agree that there are new interesting results here. I still feel that a stronger reply would have pointed to a clear conceptual difference from prior works or have provided a more compelling argument that the differences observed are of major importance. Having said this, I would tend to agree with the overall sense of the reply that the manuscript will attract substantial attention within the very active Kagomé materials area. My suggestion to the editor would, therefore, be to publish the manuscript in Nature Communications, although this is not a case where I feel strongly one way or the other.

Our Response:

We sincerely appreciate Reviewer#2 for his/her positive feedback and the overall assessment of our manuscript. His/her acknowledgment of its topicality, quality, and value is highly encouraging to us. We are grateful for his/her recognition that our work presents new and interesting results.

Regarding my numbered points:

1. This issue is mostly solved, but it looks like Figure 2d,f uses the acronym "FB" in red where they actually mean the "shadow flat band". This holds a bit more potential for confusion for readers who are not reading all the text in detail since the end of the caption says that flat bands are marked by red dashed lines. I would suggest circling the flat band in black panels d,f,&i to match the color of the arrows in panel g (of course, there is no need to use black, but it would be nice to match the colors). I would then use a colored arrow or box to mark the shadow flat band (SFB) in panels d&f and match the color to that used in the arrows of g (green may work well). The other issues are adequately addressed.

Our Response:

In response, we have revised Figure 2 by modifying the abbreviations and substituting the mention of the additional flat band with "SFB" (Shadow Flat Band). For clear visualization of these flat bands, we incorporated matching-colored circles in Fig 2d, f, and i, and denoted the position of the shadow flat band with a green arrow in Fig 2g. We would like to once again express our sincere

gratitude for the comprehensive feedback provided by Reviewer#2.

Reviewer #3 (Remarks to the Author):

The authors have addressed my comments and made substantial revisions to the text which I believe have improved the clarity of the motivation and strength of the presented arguments, especially concerning the nemacity.

As mentioned previously, the data is of very high quality. It is also fair to say that this material system already attracts a large interest in the community as evidenced by the arXiv papers which were not available at the time of original submission but are now acknowledged by the authors. With this in mind, I think a publication in Nature Communications may be warranted, however, I have two outstanding comments/suggestions:

Our Response:

We sincerely appreciate Reviewer#3 for his/her positive feedback on the revisions made to our manuscript in response to his/her comments. Additionally, we appreciate his/her recognition of the growing interest in the community regarding this material system, as evidenced by the arXiv papers that have now been acknowledged in our revised manuscript. We believe that the significance and impact of our findings, coupled with the improved clarity and alignment with community interest, make it a suitable choice for a wide readership.

- A common criticism of the manuscript among all the reviewers was the lack of direct influence of the observed features on the interesting physics such as superconductivity in this material. I think the authors' response to this is reasonable but is not very well conveyed in the revised text. For example, the comment about the expected tunability of the band structure by substitution should be emphasized more.

Our Response:

We recognize that understanding the material comprehensively requires recognizing the crucial relationship between the observed features and the underlying physics, such as superconductivity. Therefore, in this revised manuscript, we emphasize the potential impact of the observed features on the intriguing physics of the system. Specifically, we also highlight the anticipated tunability of the band structure through substitution, which can significantly influence the material's properties. The detailed revision regarding this point can be found in the list of changes.

- I appreciate the distinction that the authors make between the "flat bands" well-visible in the raw ARPES spectra and the "shadow flat bands" which appear rather as an intensity cut-off in the raw

spectra. I would like to see this distinction carried over to the figures where the shadow features are simply labelled "FB".

Our Response:

We sincerely appreciate Reviewer #3 for his/her recognition of the distinction we made between the "flat bands" and the "shadow flat bands" in the raw ARPES spectra. We agree that it would be beneficial to carry over this distinction to the figures by explicitly labeling the shadow features as "FB" to differentiate them from the prominent flat bands. In response to his/her suggestion, we have revised the Figure 2 accordingly to clearly label the shadow features to reflect their nature as intensity cut-offs in the raw spectra in the revised manuscript. Detailed changes can be found in list of changes 1-2.

REVIEWERS' COMMENTS

Reviewer #1 (Remarks to the Author):

In their answer, the authors have done exactly what I expected, which is to simulate the autocorrelation functions expected for different parts of the Fermi Surface. I think this (especially Fig. S12) will help tremendously the reader to figure out what the pictures of Fig. 5 could mean. I have to say, it's not so clear to me why the q4/p-orbital scattering would dominate so strongly the scattering. Anyway, the part due to q3 is now clearly highlighted in Fig. S15, an information that was, I think, totally missing in the previous version. I agree it could be considered as a sign of nematicity in alpha/delta scattering.

With this discussion clarified, I consider it will not be misleading to publish the manuscript. As highlighted by all referees, this work is neither extremely innovative nor giving rise to very unexpected results. However, it brings clear information on a subject of high current interest, which probably justifies publication in Nature Communications.

NB : there is a type in Fig. S3b title (oribits).

Reviewer #3 (Remarks to the Author):

The authors have addressed all my comments. I have no further suggestions to add and I recommend the manuscript for publication in Nature Communications.

Reviewer #1 (Remarks to the Author):

In their answer, the authors have done exactly what I expected, which is to simulate the autocorrelation functions expected for different parts of the Fermi Surface. I think this (especially Fig. S12) will help tremendously the reader to figure out what the pictures of Fig. 5 could mean. I have to say, it's not so clear to me why the q_4/p -orbital scattering would dominate so strongly the scattering. Anyway, the part due to q_3 is now clearly highlighted in Fig. S15, an information that was, I think, totally missing in the previous version. I agree it could be considered as a sign of nematicity in α/δ scattering.

With this discussion clarified, I consider it will not be misleading to publish the manuscript. As highlighted by all referees, this work is neither extremely innovative nor giving rise to very unexpected results. However, it brings clear information on a subject of high current interest, which probably justifies publication in Nature Communications.

NB : there is a type in Fig. S3b title (oribits).

Our response:

Thank you for your insightful feedback regarding our research paper. We appreciate your positive remarks about our efforts to simulate the autocorrelation functions for different parts of the Fermi Surface. And we will fix the typo in Supplementary Figure S3.

Thank you once again for taking the time to review our work and for providing constructive feedback.

Reviewer #3 (Remarks to the Author):

The authors have addressed all my comments. I have no further suggestions to add and I recommend the manuscript for publication in Nature Communications.

Our response:

Thank you for your thorough review of our manuscript, and we greatly appreciate your time and effort in providing valuable feedback that has helped us improve the quality of our research.